# Anxiety dissociates the adaptive functions of sensory and motor response enhancements to social threats

**Marwa El Zein[1,2]\*, Valentin Wyart[1†], Julie Grèzes[1†]**

[1]Laboratoire de Neurosciences Cognitives, Département d'Etudes Cognitives, Ecole Normale Supérieure, PSL Research University, Paris, France; [2]Université Pierre et Marie Curie, Paris, France

**Abstract** Efficient detection and reaction to negative signals in the environment is essential for survival. In social situations, these signals are often ambiguous and can imply different levels of threat for the observer, thereby making their recognition susceptible to contextual cues – such as gaze direction when judging facial displays of emotion. However, the mechanisms underlying such contextual effects remain poorly understood. By computational modeling of human behavior and electrical brain activity, we demonstrate that gaze direction enhances the perceptual sensitivity to threat-signaling emotions – anger paired with direct gaze, and fear paired with averted gaze. This effect arises simultaneously in ventral face-selective and dorsal motor cortices at 200 ms following face presentation, dissociates across individuals as a function of anxiety, and does not reflect increased attention to threat-signaling emotions. These findings reveal that threat tunes neural processing in fast, selective, yet attention-independent fashion in sensory and motor systems, for different adaptive purposes.

**\*For correspondence:**
marwaelzein@gmail.com

[†]These authors contributed equally to this work

**Competing interests:** The authors declare that no competing interests exist.

## Introduction

Perceptual decisions rely on the combination of weak and/or ambiguous samples of sensory evidence. The accuracy of this decision process is particularly important for the interpretation of negative signals, which require rapid and adaptive responses. In the social domain, identifying the emotional state of a conspecific – e.g., is he/she angry or afraid? – rarely depends solely on facial features, which are usually ambiguous and can imply different levels of threat for the observer. Surrounding cues, such as gaze direction and body posture, are known to act as contextual information during emotion recognition (*Righart and de Gelder, 2008*; *Barrett and Kensinger, 2010*; *Aviezer et al., 2011*). Specifically, the detection of anger represents an immediate threat for the observer when paired with a direct gaze; by contrast, it is when paired with an averted gaze that fear marks the presence (and possibly the localization) of a threat in the environment (*Sander et al., 2007*). These threat-signaling combinations of gaze direction and emotion have been shown to be better recognized and rated as more intense than other combinations (*Adams and Kleck, 2003*, *2005*; *Graham and LaBar, 2007*; *Sander et al., 2007*; *Bindemann et al., 2008*; *N'Diaye et al., 2009*), and this as a function of anxiety level of the individuals (*Ewbank et al., 2010*). However, the computational mechanisms underlying the prioritization of threat-signaling information remain unspecified.

Classical decision theory distinguishes two classes of mechanisms by which contextual information such as gaze direction could influence the recognition of negative emotions. Gaze direction could *bias* the interpretation of negative facial expressions in favor of the emotion signaling higher threat in this context – anger for direct gaze, fear for averted gaze. In signal detection theoretical terms

**eLife digest** Facial expressions can communicate important social signals, and understanding these signals can be essential for surviving threatening situations. Past studies have identified changes to brain activity and behavior in response to particular social threats, but it is not clear how the brain processes information from the facial expressions of others to identify these threats. Here, El Zein, Wyart and Grèzes aimed to identify how signals of threat are represented in the human brain.

The experiment used a technique called electroencephalography to record brain activity in healthy human volunteers as they examined angry and fearful facial expressions. El Zein, Wyart and Grèzes found that emotions that signaled a threat to the observer are better represented in particular regions of the brain – including those that control action – within a fraction of a second after the facial expression was shown to the volunteer. Moreover, the response of the brain regions that control action was greater in volunteers with higher levels of anxiety, which highlights the role of anxiety in reacting rapidly to social threats in the environment.

El Zein, Wyart and Grèzes' findings show that social threats can alter brain activity very rapidly, and in a more selective manner than previously believed. A future challenge is to find out whether other aspects in threatening environments can stimulate similar increases in brain activity.

(*Green and Swets, 1966*; *Macmillan and Creelman, 2004*), this effect would correspond to an additive shift of the decision criterion as a function of gaze direction. However, gaze direction could also increase the perceptual *sensitivity* to the facial features diagnostic of the emotion signaling higher threat. In contrast to the first account, this effect would correspond to a multiplicative boost of threat-signaling cues in the decision process. While the two accounts predict similar effects of gaze direction on the recognition of threat-signaling emotions, a bias effect would be maximal for neutral (emotionless) expressions, whereas a sensitivity effect would be maximal at low emotion strengths (*Figure 1*).

Here we arbitrated between these two possible accounts by recording human electroencephalogram (EEG) signals while participants categorized facial expressions as displaying anger or fear. We manipulated emotion strength by presenting 'morphed' facial expressions ranging from neutral to intense anger or fear, and contextual information by pairing facial expressions with direct or averted gaze. The parametric control over emotion strength afforded fitting decision theoretical models to the behavioral and neural data to arbitrate between bias and sensitivity accounts of threat-dependent effects on emotion recognition. At the neural level, previous studies have reported interactions between emotion and gaze direction from 200 ms following face presentation (*Sato et al., 2004*; *N'Diaye et al., 2009*; *Adams et al., 2012*; *Conty et al., 2012*), but failed to characterize the computational mechanism responsible for these effects. Here, we applied model-guided regressions of single-trial EEG signals to determine whether the neural 'encoding' of threat-signaling emotions is enhanced in ventral face-selective and/or dorsal motor regions (*El Zein et al., 2015*), and whether this enhancement is mediated by increased top-down attention to threat-signaling facial features. As high-anxious individuals show increased sensitivity to threats, but also negative signals in general (*Bishop, 2007*; *Cisler and Koster, 2010*), we further assessed the neural mechanisms by which anxiety influences the detection of and reaction to social threats.

## Results

### Behavior

Participants were presented at each trial with a face expressing fear or anger of varying emotion strength (7 levels of emotion strength for each emotion) and had to categorize the displayed emotion (*Figure 2*). Crucially, direction of gaze (direct or averted) was manipulated independently of the displayed emotion in a completely implicit fashion, as it was never mentioned to the subjects nor relevant to the emotion categorization task. Nevertheless, in addition to an expected increase in categorization performance with emotion strength ($F_{6,138} = 187.3$, $p<0.001$), gaze direction strongly

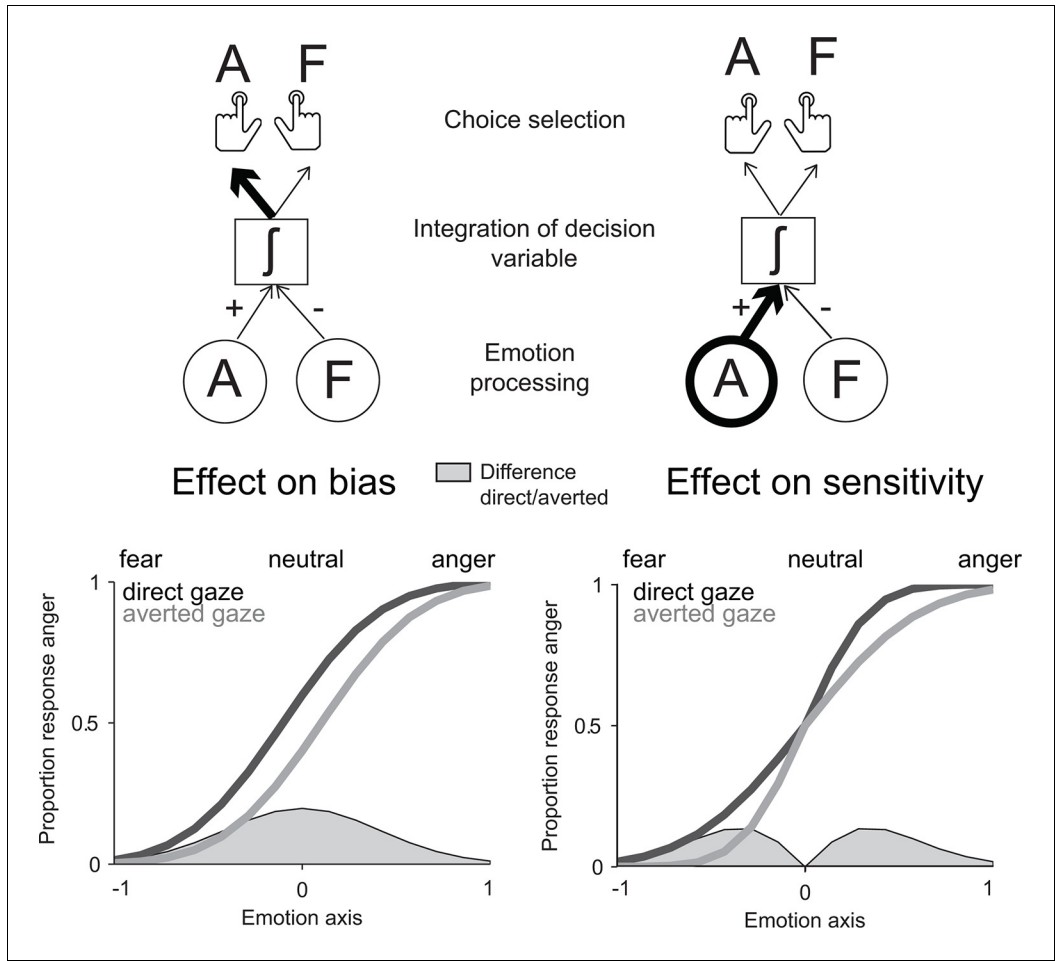

**Figure 1.** Model predictions for the effect of gaze direction on emotion categorization. Left panel: prediction of an effect of gaze direction on decision bias. Upper left panel: if gaze direction biases the interpretation of negative facial expressions in favor of the emotion signaling higher threat, direct gaze would additively bias the choice selection toward anger. Lower left panel: the predicted psychometric function would accordingly be shifted to the left for direct gaze, as participants will be biased toward interpreting faces displaying a direct gaze as angry. Maximal effects would appear for neutral (emotionless) expressions as highlighted through the filled grey area on the emotion axis that represents the difference between the two psychometric functions for direct and averted gaze. Right panel: prediction of an effect of gaze direction on perceptual sensitivity. Upper right panel: if gaze direction increases the sensitivity to the facial features diagnostic of the emotion signaling higher threat, direct gaze would now multiplicatively boost the processing of an angry expression displaying a direct gaze. Lower right panel: the predicted psychometric function would now show an increased slope for threat-signaling emotions, with maximal effects at low emotion strengths (as shown in the filled grey area on the emotion axis).

interacted with the displayed emotion on performance ($F_{1,23} = 21.2$, $p<0.001$). Facial displays of anger were better categorized when paired with a direct gaze ($t_{23} = 4.3$, $p<0.001$), whereas expressions of fear were better categorized when paired with an averted gaze ($t_{23} = -3.4$, $p<0.01$; *Figure 3a*). These combinations of gaze and emotion, anger paired with a direct gaze and fear paired with an averted gaze, are associated with higher threat for the observer (*Sander et al., 2007*), albeit of different natures. In the case of anger, gaze direction indicates the target of the threat, while in the case of fear gaze direction signals its source. Nevertheless, just as the combination of anger with a direct gaze is more threatening/relevant than with an averted gaze, fear is more threatening when paired with an averted gaze than with a direct gaze. These two combinations, anger direct and fear averted, will thus be labeled as THREAT+ combinations as opposed to THREAT− combinations (i.e., anger paired with averted gaze, and fear paired with direct gaze).

Moreover, a significant emotion by gaze by emotion strength interaction was observed ($F_{6,138} = 4.3$, $p<0.01$), explained by a stronger influence of gaze on emotion categorization at weak emotion strengths (gaze by emotion interaction for levels 1 to 4, $F_{1,23} = 23.8$, $p<0.001$) than at high emotion

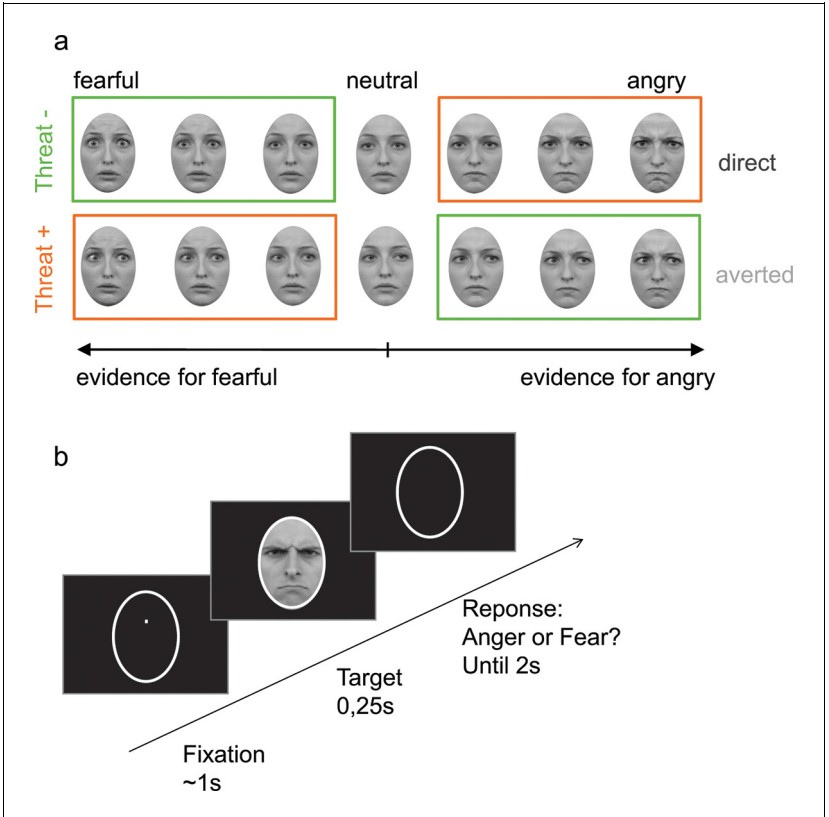

**Figure 2.** Stimuli and experimental procedure. (a) Examples of morphed expressions for one identity: morphs from neutral to intense fearful/angry expressions providing evidence for one or the other emotion. Stimuli displayed either an averted or a direct gaze. THREAT+ conditions (in orange) correspond to combinations of gaze and emotion that signal higher threat for the observer as compared to THREAT− conditions (in green). (b) Following fixation, a facial expression appeared for 250 ms, after which the participant had to indicate whether the face expressed anger or fear within 2 seconds. No feedback was provided after response.

strengths (gaze by emotion interaction for levels 5 to 7, $F_{1,23} = 5.1$, p<0.05). Reaction time (RT) analyses revealed a decrease of correct RTs with emotion strength (repeated-measures ANOVA, $F_{6,138} = 54.5$, p<0.001), faster responses to angry as compared fearful faces ($F_{1,23} = 12$, p<0.01), and faster responses to direct as compared to averted gaze ($F_{1,23} = 7.7$, p<0.05). Furthermore, an emotion by gaze interaction was observed ($F_{1,23} = 8$, p<0.01), corresponding to faster reaction times for direct as compared to averted gaze in the anger condition only ($t_{23} = -3.9$, p<0.001).

To characterize the mechanism underlying the improved recognition of threat-signaling emotions, we fitted participants' behavior using a family of nested models of choice which hypothesize that decisions are formed on the basis of a noisy comparison between the displayed emotion and a criterion, under the following formulation (see Materials and methods for details):

$$P(anger) = \Phi[w \cdot x + b] \cdot (1 - \varepsilon) + 0.5 \cdot \varepsilon$$

where P(anger) corresponds to the probability of judging the face as angry, $\Phi[.]$ to the cumulative normal function, *w* to the perceptual sensitivity to the displayed emotion, **x** to the evidence (emotion strength) in favor of anger or fear in each trial (from -7 for an intense expression of fear to +7 for an intense expression of anger), *b* to an additive stimulus-independent bias in favor one of the two responses/emotions, and $\varepsilon$ to the proportion of lapses (random guesses) across trials.

We compared a 'null' model which did not allow for contextual influences of gaze direction on the decision process, to two additional models which instantiate two different mechanisms which could account for the observed increase in recognition accuracy for THREAT+ combinations of gaze and emotion: 1. a first variant in which gaze direction *biases* the decision criterion in favor of the

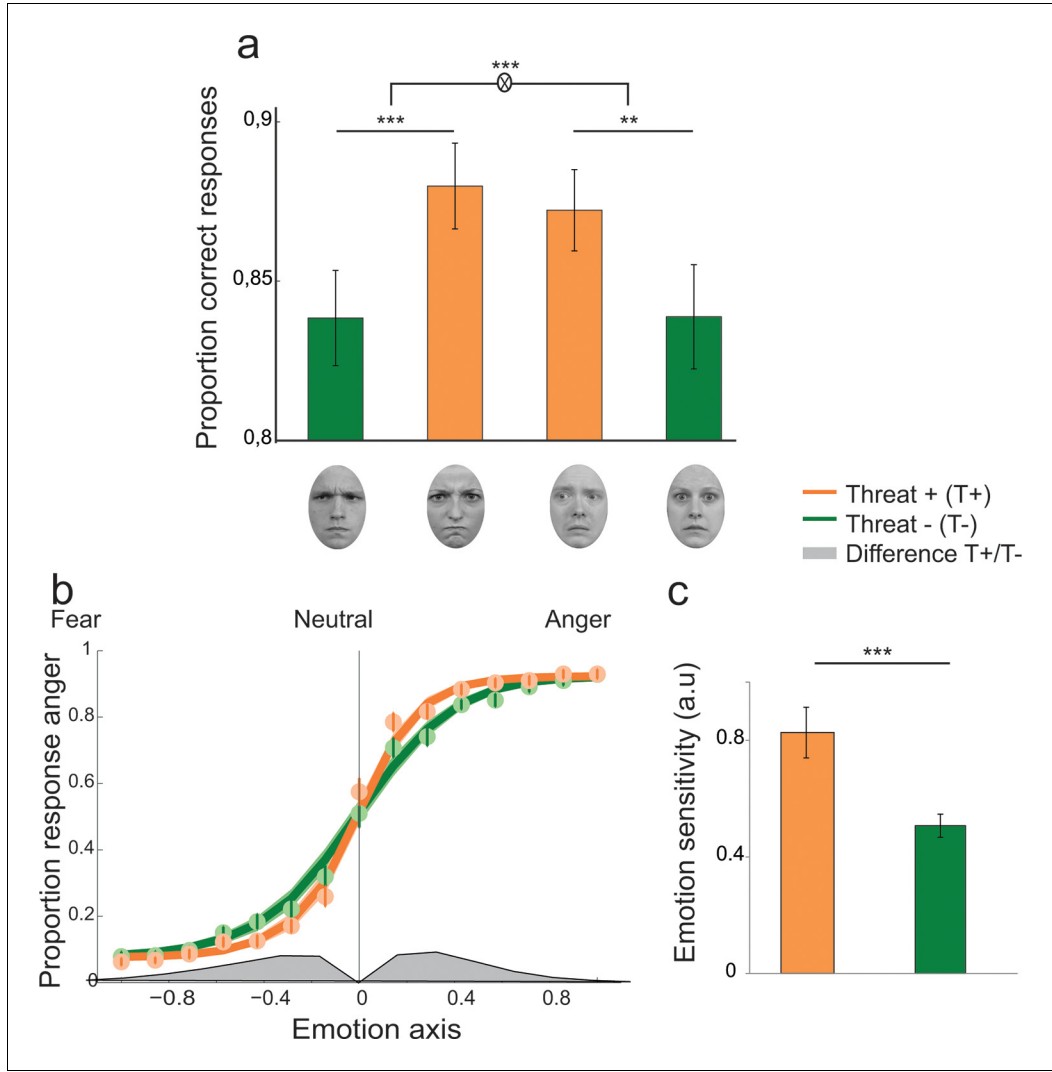

**Figure 3.** Enhanced recognition accuracy and perceptual sensitivity to threat-signaling emotions. (a) Proportion of correct responses for (from left to right) averted/anger, direct/anger, averted/fear and direct/fear. THREAT+ combinations of gaze and emotion (in orange) were associated with increased recognition accuracy. (b) Psychometric function representing the proportion of 'anger' responses as a function of the evidence for anger (proportion morph, 0 = neutral, negative towards fear, and positive towards anger) for THREAT+ (orange) and THREAT− (green) combinations of gaze and emotion. Dots and attached error bars indicate the human data (mean ± s.e.m.). Lines and shaded error bars indicate the predictions of the best-fitting model. (c) Parameter estimate for the slope of the psychometric curve (corresponding to emotion sensitivity) for THREAT+ and THREAT− combinations. **p < 0.01, ***p < 0.001.

emotion signaling higher threat, and 2. a second variant in which gaze direction enhances the *sensitivity* to the emotion signaling higher threat. Bayesian model selection revealed that a sensitivity enhancement for THREAT+ combinations explained substantially better the behavioral data than a criterion shift (Bayes Factor $\approx 10^8$, exceedance probability $p_{exc} > 0.74$). Maximum-likelihood estimates of the perceptual sensitivity parameter $w$ extracted from the winning model were significantly increased for THREAT+ combinations of gaze and emotion ($t_{23} = 3.9$, p<0.001; *Figure 3b,c*). The proportion of lapses did not vary between THREAT+ and THREAT- combinations ($t_{23} = 0.4$, p>0.5).

## Enhanced neural encoding of threat-signaling emotions

To validate the finding of enhanced sensitivity to threat-signaling emotions, and to identify its neural substrates, we then investigated how facial expressions modulated scalp EEG activity recorded during the emotion categorization task. Instead of computing event-related averages, we relied on a

parametric regression-based approach consisting in regressing single-trial EEG signals against the strength of the displayed emotion at each electrode and time point following the presentation of the face (*Wyart et al., 2012a, 2015*). A general linear regression model (GLM) was fit to the EEG data where emotion strength (from 0 for a neutral/emotionless expression to 7 for an intense fear/ anger expression) was introduced as a trial-per-trial predictor of broadband EEG signals at each electrode and time point following stimulus onset (from 0.2 s before to 1.0 s following stimulus onset). The resulting time course at each electrode represents the degree to which EEG activity 'encodes' (co-varies with) the emotion strength provided by morphed facial features.

Parameter estimates of the regression slope revealed significant correlations between emotion strength and EEG activity peaking initially around 280 ms following face presentation at temporal (t-test against zero, $t_{23}$ = -12.7, p<0.001) and frontal electrodes ($t_{23}$ = 8.7, p<0.001), and then around 500 ms and at response time at centro-parietal ($t_{23}$ = 10.2, p<0.001) and frontal electrodes ($t_{23}$ = -7.9, p<0.001) (*Figure 4a–c*). Time points and electrodes where parameter estimates diverge significantly from zero indicate neural encoding of emotion information. The strength of this neural encoding – indexed by the amplitude of the parameter estimate – provides a measure of the *neural sensitivity* to emotion information.

To test for a neural signature of the increased sensitivity to threat-signaling emotions, we compared parameter estimates extracted separately for THREAT+ (anger direct and fear averted) and THREAT− (anger averted and fear direct) combinations of gaze and emotion. This contrast revealed increased parameter estimates for THREAT+ combinations first at 170 ms at temporal (paired t-test, $t_{23}$ = -2.5, p<0.05) and frontal electrodes ($t_{23}$ = 2.2, p<0.05), and then later at 500 ms and at response time at centro-parietal ($t_{23}$= 2.2, p<0.05) and frontal electrodes ($t_{23}$ = -2.4, p<0.05) (*Figure 4a–c*). This finding indicates that the neural gain of emotion encoding was enhanced at these time points and electrodes for threat-signaling emotions. This threat-dependent enhancement remained significant when considering only correct responses (temporal: $t_{23}$ = -2.1, p<0.05; centro-parietal $t_{23}$ = 4.2, p<0.001). Interestingly, THREAT+ combinations were not associated with increased event-related averages at classical peak latencies (P1, N170, P2, P3: all $t_{23}$ < 1.95, p>0.07). To assess which brain regions generated the scalp-recorded EEG signals, we computed the cortical sources of this enhanced encoding of threat-signaling emotions by performing the same regression approach to minimum-norm current estimates distributed across the cortical surface. Parameter estimates at time points of interest (where differences between THREAT+ and THREAT− combinations were observed) were then contrasted between the two conditions (see Materials and methods). Increases in regression slopes for THREAT+ combinations shifted from ventral visual areas selective to facial expressions of emotion (fusiform gyrus and superior temporal sulcus) around 170 ms, to associative brain regions encompassing parietal, temporal and frontal cortices (superior and middle temporal, temporal pole, and orbitofrontal cortices) at 500 ms, and then to sensorimotor regions around response onset (dorsal central, parietal and frontal regions) (*Figure 4d–f*).

These neural effects converge with behavioral modeling in favor of a sustained enhancement of perceptual sensitivity to threat-signaling emotions, starting 170 ms following face presentation and lasting until after response onset. Additional evidence supports our hypothesis that enhancements in neural sensitivity to THREAT+ combinations are specifically linked to an increase in implied threat for these combinations of gaze and emotion. A separate group of participants rated the identities used in the emotion categorization task in terms of perceived threat and trustworthiness (see Materials and methods), and the group-level ratings for each identity were regressed against single-trial EEG signals as additional regressors. This regression showed that perceived threat, but not trustworthiness, correlated significantly with temporal and centro-parietal EEG activity at 500 ms following face presentation, in the same direction as the contrast between THREAT+ and THREAT− combinations (threat: $t_{23}$ > 3.6, p<0.01; trustworthiness: $t_{23}$ < 0.7, p>0.48).

## Attention-independent enhancement of neural processing by threat

Analyses of the neural data have so far confirmed the hypothesis that contextual gaze information affects emotion categorization by increasing the perceptual sensitivity to threat-signaling emotions. Such an effect could be mediated by increased top-down attention to threat-signaling emotions – i.e., THREAT+ combinations (anger direct and fear averted). To test this possibility, we explored whether residual fluctuations in single-trial EEG signals unexplained by variations in emotion

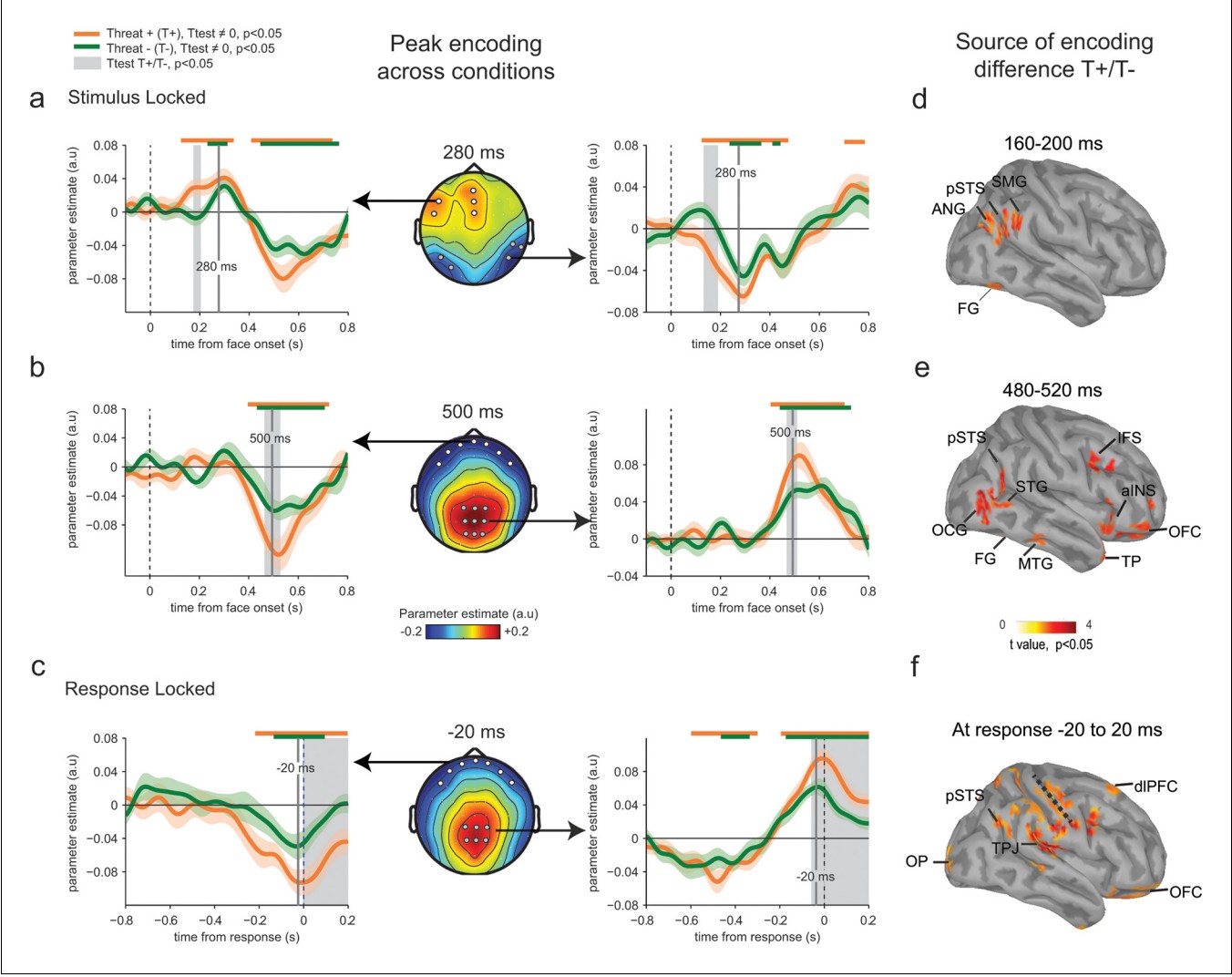

**Figure 4.** Enhanced neural encoding of threat-signaling emotions. (**a**) Middle panel: scalp topography of neural encoding at 280 ms, corresponding to its first peak of the encoding of emotion strength averaged across conditions (peak of deviation from zero), and expressed as mean parameter estimates in arbitrary units (a.u.). Dots indicate electrodes of interest where neural encoding was maximal. Left and right panels: encoding time course for THREAT+ and THREAT− conditions at electrodes of interest. Shaded error bars indicate s.e.m. Thick orange and green lines indicate significance against zero at a cluster-corrected p-value of 0.05. Shaded grey areas indicate significant differences between THREAT+ and THREAT− conditions at p < 0.05. (**b**) Same conventions as (**a**) at the second neural encoding peak at 500 ms. (**c**) Same conventions as (**a**) at the third neural encoding peak at response time. (**d**) Estimated cortical sources of the encoding difference between THREAT+ and THREAT− conditions at the time of significant difference between conditions at 170 ms. (**e**) Same as (**d**) at 500 ms. (**f**) Same as (**d**) at response time. FG: fusiform gyrus, pSTS: posterior superior temporal sulcus, SMG: supramarginal gyrus, ANG: angular gyrus, STG: superior temporal gyrus, MTG: middle temporal gyrus, OCG: occipital gyrus, aINS: anterior insula, IFS: inferior frontal sulcus, TP: temporal pole, OFC: orbitofrontal cortex, OP: occipital pole, TPJ: temporo-parietal junction, dlPFC: dorsolateral prefrontal cortex.

strength (measured by the previous regressions) modulated the accuracy of the subsequent categorical decision – i.e., the perceptual sensitivity to the displayed emotion. This approach is reminiscent of 'choice probability' measures applied in electrophysiology to measure correlations between neural activity and choice behavior (*Britten et al., 1996*; *Shadlen et al., 1996*; *Parker and Newsome, 1998*) – by estimating how much fluctuations in recorded neural signals are 'read out' by the subsequent decision (*Wyart et al., 2012a*, *2015*). Stimulus-independent improvements in neural-choice correlations have been classically interpreted as increases in 'read-out' weights – i.e., increased top-down attention to these neural signals (*Nienborg and Cumming, 2009*, *2010*). Here, an increased neural modulation of choice for THREAT+ conditions could indicate an increase in

top-down attention to threat-signaling emotions, which could in turn explain the observed increase in perceptual and neural sensitivity to these combinations of gaze and emotion.

To test this hypothesis, we entered EEG residuals from the previous regression against emotion strength as an additional 'mediation' predictor of choice – as means to test whether these neural signals co-vary with perceptual sensitivity (see Materials and methods for details). In practice, we estimated the parameters $b_{mod}$ and $w_{mod}$ of these neural modulation terms at each time point following face presentation via an EEG-informed regression of choice for which the trial-by-trial neural residuals $\mathbf{e}$ from the regression against emotion strength were entered either alone (additive influence, parameter $b_{mod}$) or as their interaction with emotion strength (multiplicative influence, parameter $w_{mod}$) as additional predictors of the subsequent choice:

$$P(anger) = \phi[(w + w_{mod} \cdot e)\, x + b + b_{mod} \cdot e]$$

The time course and spatial distribution of this neural modulation of perceptual sensitivity (*wmod*) followed qualitatively the neural encoding of emotion strength (*Figure 5a–c*), with a negative temporal component peaking at 270 ms ($t_{23}$ = -4.2, p<0.001), followed by a positive centro-parietal one peaking around 600 ms ($t_{23}$ = 8.0, p<0.001) and then at response time ($t_{23}$ = 7.6, p<0.001). We used Bayesian model selection to confirm that EEG residuals co-varied multiplicatively with the perceptual sensitivity (*wmod*) of the subsequent decision, not additively as a bias (*bmod*) in emotion strength, both at temporal (Bayes factor ≈ $10^{3.4}$, $p_{exc}$ = 0.79) and centro-parietal electrodes (Bayes factor ≈ $10^{8.9}$, $p_{exc}$ = 0.99). Critically, no difference in modulation strength was observed between THREAT+ (anger direct and fear averted) and THREAT− (anger averted and fear direct) combinations (temporal: $t_{23}$ = -0.4, p>0.5; centro-parietal: $t_{23}$ = 0.1, p>0.5). To determine whether this absence of significant difference is due to a genuine absence of effect (rather than a lack of statistical sensitivity), we computed Bayes factors under the same parametric assumptions as conventional statistics (see Materials and methods). We obtained Bayes factors lower than 10–4 at temporal and centro-parietal electrodes, indicative of no increase in 'read-out' weights for THREAT+ conditions. This null effect suggests that the observed enhancement in perceptual and neural sensitivity to these threat-signaling combinations of gaze and emotion is not triggered indirectly by an increase in top-down attention in these conditions.

## Early neural encoding of threat-signaling emotions in motor preparation

We reasoned that threat could impact not only the neural representation of the displayed emotion in visual and associative cortices, but also the preparation of the upcoming response in effector-selective structures (*Conty et al., 2012*). To measure response-preparatory signals in the neural data, we computed spectral power in the mu and beta frequency bands (8–32 Hz) (*Donner et al., 2009*; *de Lange et al., 2013*). Limb movement execution and preparation coincide with suppression of low-frequency (8–32 Hz) activity that is stronger in the motor cortex contralateral as compared to ipsilateral to the movement. Thus, subtracting the contralateral from ipsilateral motor cortex activity is expected to result in a positive measure of motor preparation. The contrast between left-handed and right-handed responses at response time identified lateral central electrodes, associated with focal sources in motor cortex (*Figure 6a*). Subtracting contralateral from ipsilateral signals relative to the hand assigned to the 'fear' response (counterbalanced across participants) provided a motor lateralization index whose sign predicts significantly the upcoming choice (anger or fear) from 360 ms before response onset (paired t-test, $t_{23}$ = 4.6, p<0.001; *Figure 6b*).

We applied the previous neural encoding approach by regressing this motor lateralization index against the *signed* emotion strength (from 0 for a neutral expression, to ±7 for an intense anger/fear expression) on a trial-by-trial basis. Parameter estimates of the regression slope diverged significantly from zero from 400 ms after stimulus onset (t-test against zero, $t_{23}$ = 5.1, p<0.001) and at response time ($t_{23}$ = 5.2, p<0.001) – reflecting stronger response preparation to stronger (i.e., more diagnostic) emotions. Computing regression slopes separately for THREAT+ (anger direct and fear averted) and THREAT- (anger averted and fear direct) combinations revealed that THREAT+ combinations produced a stronger encoding of emotion strength in motor preparation late at response onset ($t_{23}$ = 2.9, p<0.01), but also early around 200 ms following face presentation ($t_{23}$ = 3.2, p<0.01). This early threat-dependent motor enhancement remained significant when considering only correct responses ($t_{23}$ = 3.0, p<0.01). While THREAT− combinations of gaze and emotion were

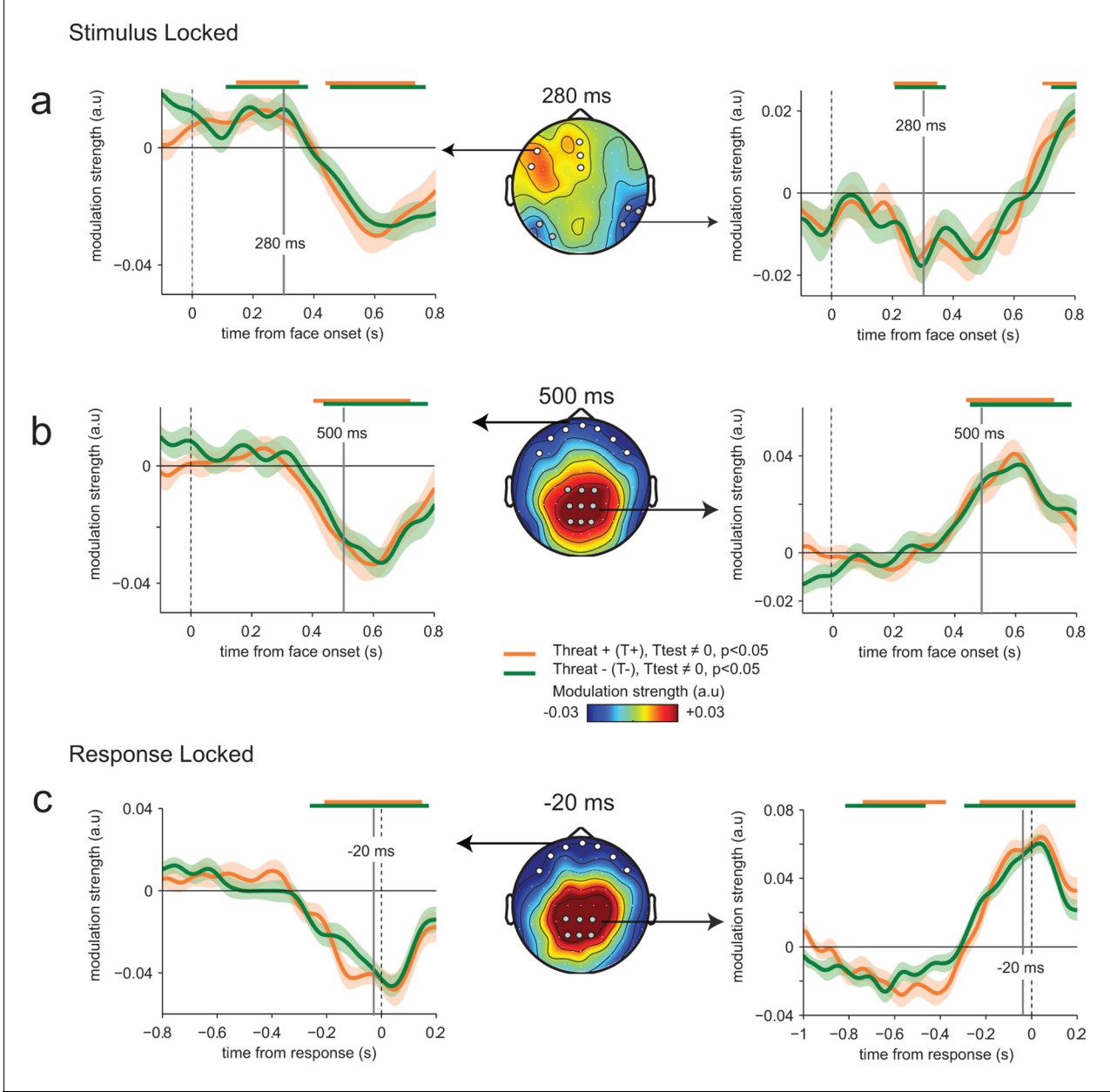

**Figure 5.** Absence of threat-dependent enhancement of neural-choice correlations. (a) Middle panel: scalp topography of neural-choice correlations, expressed as the modulation of perceptual sensitivity by EEG encoding residuals at 280 ms, same time point shown in *Figure 4a*. Electrodes of interest indicated with dots are the same as in *Figure 4a*. Left and right panels, time course of the modulation of perceptual sensitivity by EEG encoding residuals expressed in arbitrary units (a.u.). Same conventions as in *Figure 4a*. (b) Same conventions as (a) at 500 ms. (c) Same conventions as (a) at response time. The variation of the modulation strength over time is consistent with the variation of the encoding parameter estimate. No difference between THREAT+ and THREAT− is observed.

not associated with significant neural encoding in motor preparation until 440 ms following face presentation ($t_{23} < 0.8$, $p > 0.4$), THREAT+ combinations resulted in significant neural encoding between 100 and 320 ms, peaking at 200 ms ($t_{23} = 3.2$, $p < 0.01$; *Figure 6c*).

To determine whether this early neural encoding of threat-signaling emotions in motor preparation influences the speed of subsequent responses, we recomputed and compared regression parameters estimated separately for fast and slow responses to THREAT+ combinations (anger direct and fear averted), on the basis of a median split of response times informed by emotion

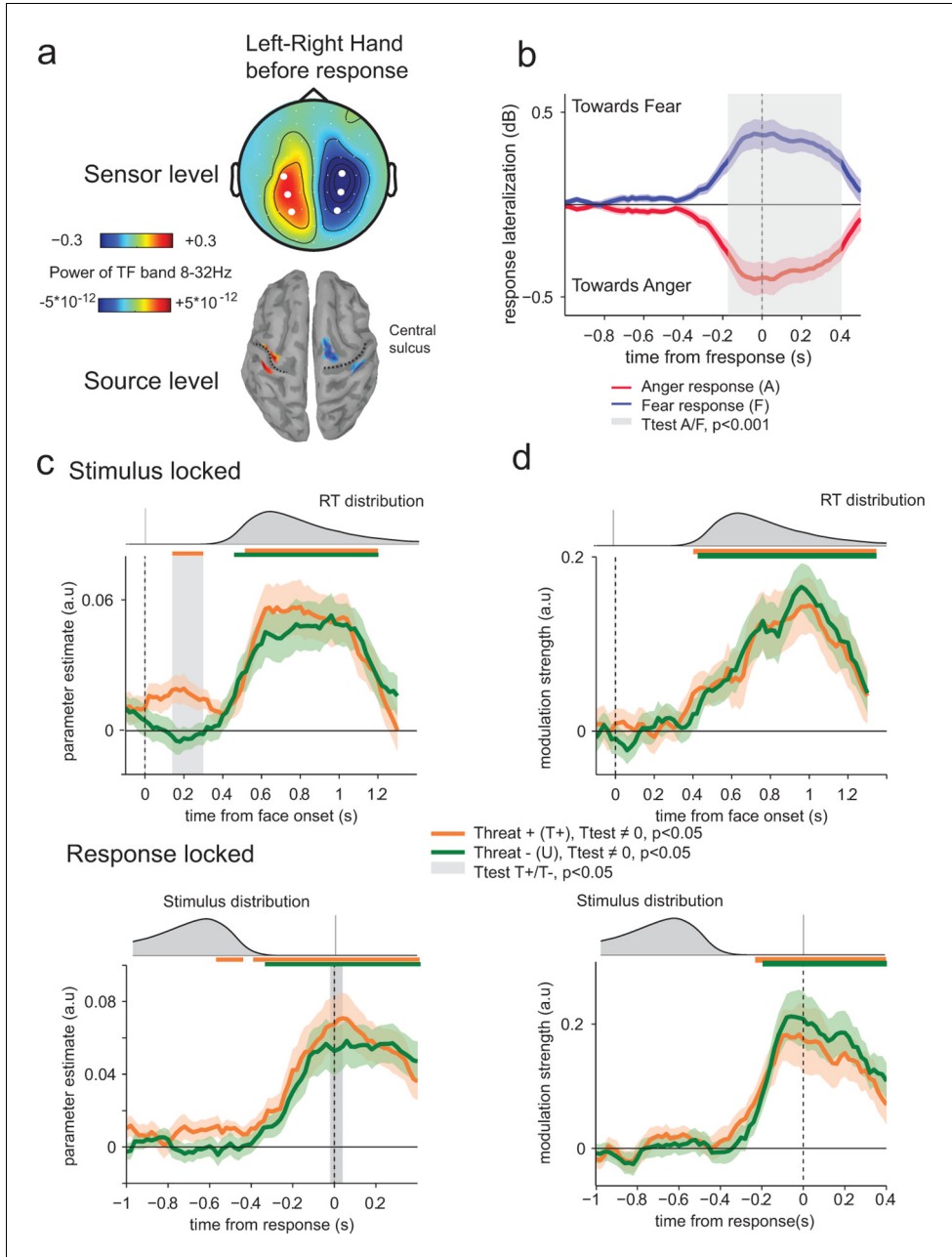

**Figure 6.** Encoding of threat-signaling emotions in motor response lateralization measures. (a) Top panel, scalp topography before response of the time frequency power in the 8–32 Hz band in the last 100 ms before response, for the trials where subjects responded with their left hand minus the trials where they responded with their right hand. Dots correspond to the selected electrodes, where the effect was maximal. Bottom panel: corresponding neural sources. (b) Time course of response lateralization (time frequency power activity from the contralateral electrodes minus ipsilateral electrodes to the hand used to respond 'fear') towards anger and fear when the choice was anger (red) or fear (blue). Shaded error bars indicate s.e.m. The shaded gray area indicates a significant difference in motor lateralization between Anger and Fear responses. (c) Encoding of emotion strength in response lateralization index for THREAT+ (orange) and THREAT− (green) conditions. Differences between conditions are observed at 200 ms after stimulus onset (stimulus-locked, upper panel) and at response time (response-locked, lower panel). Conventions are the same as in *Figure 4*. (d) Time course of neural-choice correlations, expressed as the modulation of additive bias by motor lateralization encoding residuals in arbitrary units (a.u.) stimulus-locked (upper panel) and response locked (lower panel). Conventions are the same as in *Figure 4*.

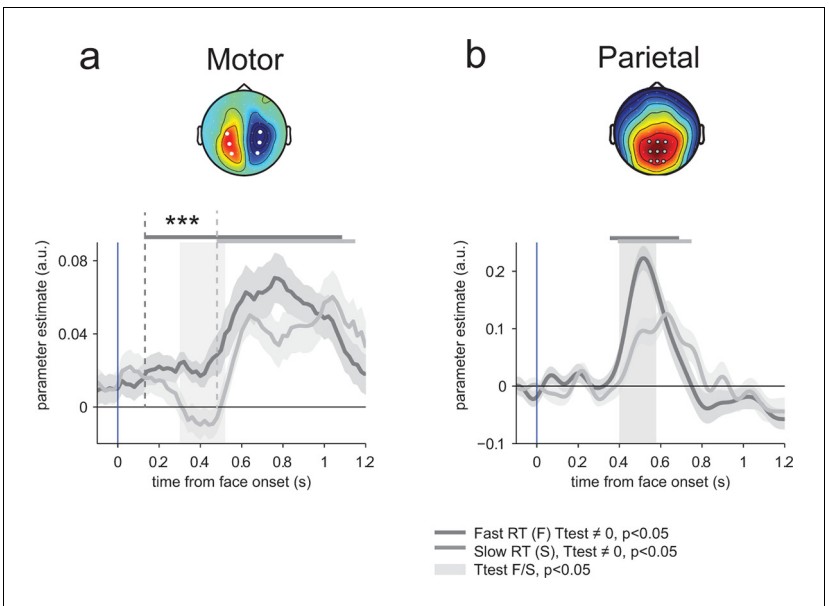

**Figure 7.** Encoding of emotion strength as a function of reaction times (RT) in motor and parietal structures. (a) Neural encoding of emotion strength for THREAT+ conditions in motor lateralization for fast and slow reaction times (RT): when RTs were fast, the encoding of emotion strength became significant at 150 ms and rose gradually until response; by contrast, when RTs were slow, the encoding of emotion strength became significant later at 540 ms. Shaded error bars indicate s.e.m. Thick dark and light grey lines indicate significance against zero at a cluster-corrected p-value of 0.05. Shaded grey bars indicate significant differences between fast and slow responses. Encoding latency is significantly different between fast and slow RTs, ***: p<0.001 (b) Emotion strength encoding in parietal electrodes. Convention are the same than (a). Fast responses are associated with a stronger neural encoding of emotion strength, but without any change in encoding latency.

strength. This comparison revealed a single, gradual neural encoding of emotion strength in motor preparation preceding fast, but not slow responses, arising as early as 150 ms (at a threshold p-value of 0.05) following the presentation of the face (difference in encoding onset between fast and slow responses, jackknifed (*Kiesel et al., 2008*, see Materials and methods) $t_{23}$ = 5.2, p<0.001; *Figure 7a*). This effect indicates that the early neural encoding of THREAT+ combinations in motor preparation is characteristic of efficient (fast) responses. We verified that this latency shift in neural encoding was selective of motor preparation signals, by performing the same comparison on the neural encoding of emotion strength at centro-parietal electrodes. This contrast revealed only a difference in peak amplitude, not onset latency, between fast and slow responses (peak amplitude: $t_{23}$ = 5.1, p<0.001; onset latency: jackknifed $t_{23}$ = -1.3, p>0.2; *Figure 7b*).

Finally, we performed neural-choice correlations analyses to assess whether the early neural encoding of threat-signaling emotions in motor preparation influences not only the speed, but also the content (anger or fear) of subsequent responses. Across conditions, the neural 'mediation' analysis described above revealed that stimulus-independent fluctuations in motor lateralization index covary as an additive choice bias in the upcoming response from 400 ms following face presentation ($t_{23}$ = 2.9, p<0.01). Indeed, in contrast to fluctuations in temporal and centro-parietal activity, the impact of variability in motor lateralization on emotion categorization was better described as an additive choice *bias* rather than a change in perceptual *sensitivity* (Bayes factor ≈ $10^{36.4}$, $p_{exc}$ = 0.98) – consistent with its hypothesized role as a motor representation of the decision variable (*Donner et al., 2009*; *de Lange et al., 2013*). No difference in modulation strength was observed between THREAT+ (anger direct and fear averted) and THREAT− (anger averted and fear direct) combinations ($t_{23}$ < 1.6, p>0.1; *Figure 6d*). Critically, even when considering combinations alone, residual variability in motor lateralization measured between 100 and 320 ms (where the neural encoding of threat-signaling emotions was significant) did not bias significantly the upcoming choice ($t_{23}$ < 1.4, p>0.17). This null effect was supported by Bayesian model selection that identified a

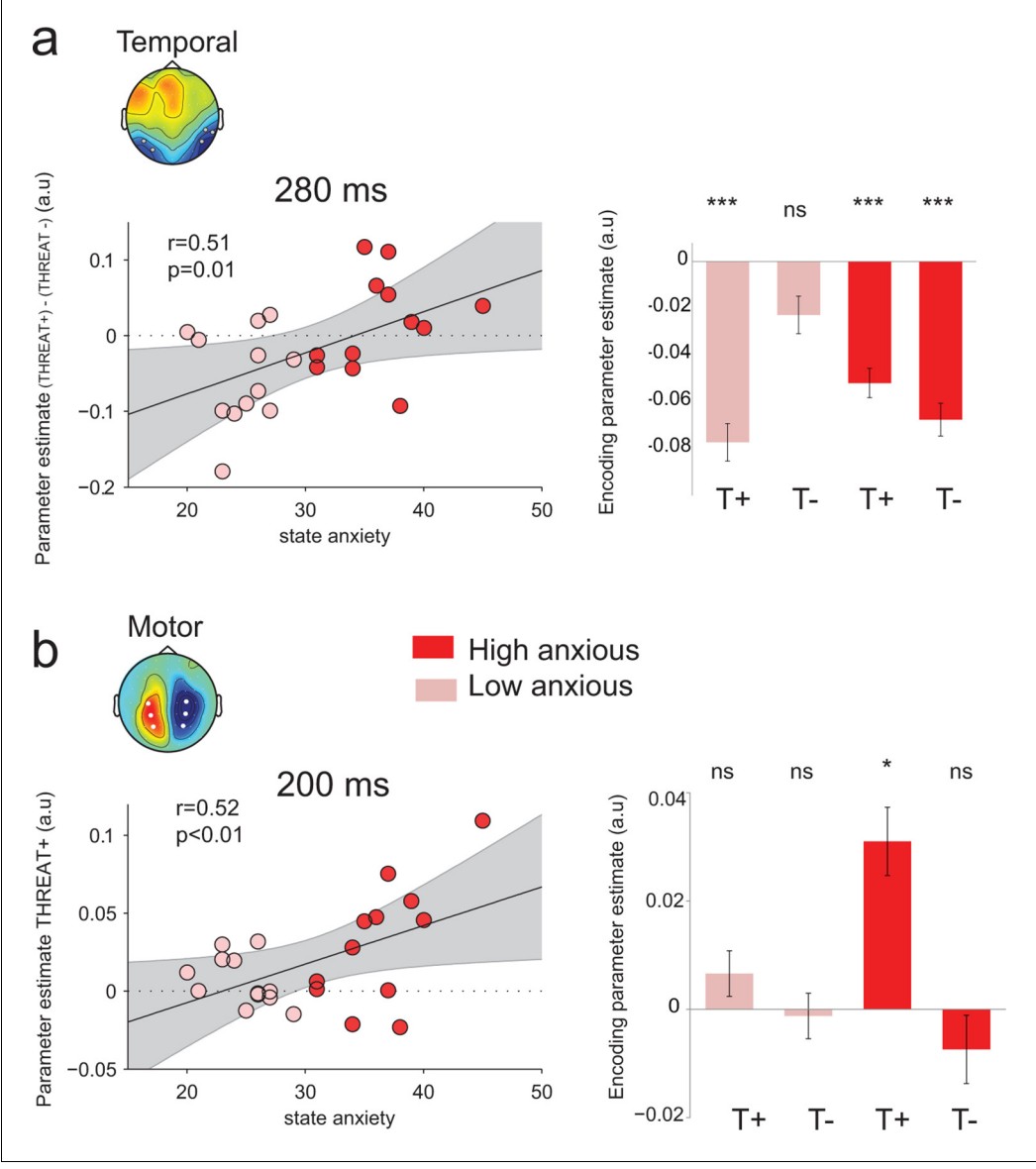

**Figure 8.** Modulation of threat encoding by individual anxiety. (**a**) Left panel: correlation (Pearson) between state anxiety and the difference of the encoding parameter estimates between THREAT+ and THREAT− conditions in temporal electrodes at 280 ms. Right panel: encoding parameter estimates in temporal electrodes split into high and low anxious individuals for both THREAT+ and THREAT− conditions at 280 ms. T+: THREAT+, T-: THREAT-. (**b**) Left, correlation (Pearson) between state anxiety and the encoding parameter estimates in motor lateralization signals for THREAT+ condition at 200 ms. Right, encoding parameter estimates in motor lateralization signals split into high and low anxious individuals for both THREAT+ and THREAT− conditions at 200 ms. ***: p<0.001, *p<0.05.

genuine absence of neural-choice correlation as the most likely account of the data (Bayes factor $\approx$ $10^{2.3}$, $p_{exc} = 0.96$). This finding indicates that the early neural encoding of threat-signaling emotions in motor preparation occurs earlier than the formation of the upcoming choice.

## Anxiety-dependent neural encoding of threat-signaling emotions

In the general population, anxiety has been classically associated with an oversensitivity to threat signals in social conditions (*Bishop, 2007*; *Cisler and Koster, 2010*). Here, we assessed whether the enhanced neural processing of threat-signaling emotions in temporal and motor regions co-varied with the level of anxiety in our participants. For this purpose, we measured anxiety at the beginning of the experimental session, before data collection, using the Spielberger State-Trait Anxiety

Inventory (STAI) (*Spielberger et al. 1983*). This self-questionnaire provides a measure of vulnerability for anxiety disorders (*Grupe et al. 2013*). Participants' state anxiety scores ranged from 20 to 45 (mean = 30.5, SD = 6.8). Trait anxiety scores ranged from 22 to 52 (mean = 38.2, SD = 7.8). These scores are comparable to the original published norms for this age group (*Spielberger, 1983*) and to those from French normative data (*Bruchon-Schweitzer and Paulhan, 1993*). We analyzed the effect of anxiety on the behavioral and neural data in two complementary ways: 1. by splitting the participants in two equally-sized groups based on their measured anxiety, and 2. by correlating neural encoding parameters estimated at the level of individual participants with their measured anxiety. Surprisingly, we found no effect of anxiety on overall measures of performance ($t_{11} < 0.05$, $p > 0.9$), nor on the difference between THREAT+ (anger direct and fear averted) and THREAT- (anger averted and fear direct) combinations of gaze and emotion ($F_{1,22} < 0.4$, $p > 0.5$).

Nevertheless, the absence of effect of anxiety at the behavioral level was accompanied by a compensatory double dissociation in the neural data. Indeed, state anxiety influenced significantly the neural encoding of emotion strength at temporal electrodes at the peak of neural encoding, 280 ms following face presentation (median split, interaction: $F_{1,22} = 7.3$, $p = 0.01$; *Figure 8a*): high-anxious observers showed no difference in neural encoding between THREAT+ and THREAT- combinations (THREAT+ : $t_{11} = -5.8$, $p < 0.001$; THREAT-: $t_{11} = -6.1$, $p < 0.001$, difference: $t_{22} = 0.84$, $p = 0.4$), whereas low-anxious observers encoded exclusively THREAT+ at the same latency (THREAT+: $t_{11} = -6.5$, $p < 0.001$; THREAT-: $t_{11} = -1.8$, $p = 0.08$; difference: $t_{22} = -3.0$, $p = 0.01$). A parametric assessment of the relationship between state anxiety and the difference in neural encoding between THREAT+ and THREAT- combinations proved to be significant (Pearson correlation coefficient r = 0.51, d.f. = 22, $p = 0.01$; *Figure 8a*). In other words, high anxiety was associated with a significant and indifferent neural encoding of negative emotions, whether threat-signaling or not, in ventral face-selective regions.

Interestingly, at the early time window (peak of the encoding at 200 ms) where only THREAT+ combinations (anger direct and fear averted) were encoded in motor signals, a reverse pattern was observed: only high anxious individuals showed a significant encoding at this latency (interaction between between-subject state anxiety and gaze pairing $F_{1,22} = 4$, $p = 0.05$; *Figure 8b*). The more the individuals were anxious, the more they encoded observer-relevant threat signals in motor systems (correlation between parameter estimates for THREAT+ conditions and state anxiety Pearson coefficient r = 0.52, d.f. = 22, $p < 0.01$; *Figure 8b*). Moreover, the neural encoding of THREAT+ emotions in motor signals correlated with behavioral sensitivity to THREAT+ emotions for high-anxious individuals (Pearson correlation coefficient r = 0.66, d.f. = 10, $p = 0.01$), whereas it was not the case for low-anxious individuals (Pearson correlation coefficient r = −0.42, d.f. = 10, $p > 0.16$, difference between coefficients, $p < 0.01$). To sum up, while high anxious individuals process all threat signals equivalently in face selective regions, they selectively encode threat signals that are relevant to them in motor specific systems, and this encoding reflects their behavioral sensitivity to threat-signaling emotions.

## Discussion

Accurate decoding of emotions in others, especially negative ones, conveys adaptive advantages in social environments. Although typical social interactions do not require an explicit categorization of the emotion expressed by others, a precise understanding of the neural mechanisms involved in emotion recognition provides important information regarding how the human brain processes socially meaningful signals. And while past work has uncovered the neural correlates of perceptual decisions (*Gold and Shadlen, 2007*; *Heekeren et al., 2008*), only few studies have addressed the issue of how such decisions are formed on the basis of socially relevant stimuli such as facial displays of emotion. As in most perceptual categorization tasks, we manipulated the ambiguity of sensory evidence – here, using controlled morphs between angry or fearful expressions and neutral ones. But owing to the social nature of our stimuli, we could simultaneously and implicitly manipulate the contextual significance of the displayed emotion in terms of implied threat for the observer, using gaze direction, and apply a model-guided approach to characterize the neural prioritization of threat-signaling information in electrical brain signals.

Gaze direction, which acts as a contextual cue in our emotion categorization task, differs from contextual cues found in perceptual decision-making studies which are typically provided hundreds

of milliseconds before the decision-relevant stimulus (*Rahnev et al., 2011*; *Kok et al., 2012*; *Wyart et al., 2012b*; *de Lange et al., 2013*). Here, as in many social situations, contextual cues can co-occur with the decision-relevant stimulus – a property which strongly constrains their impact on stimulus processing. Moreover, the meaning of contextual cues (e.g., attention or expectation cues) used in perceptual decision-making studies is usually instructed explicitly, and thus processed explicitly by the participants during task execution (*Kok et al., 2012*; *Wyart et al., 2012b*). Here, by contrast, gaze direction is irrelevant for the emotion categorization task, and thus does not need to be processed explicitly. Despite these two differences with other contextual cues, we show that gaze direction tunes the neural processing of emotion information from 200 ms following stimulus onset until response in sensory, associative and motor circuits of the human brain.

Previous observations of increased subjective ratings and improved recognition of angry expressions paired with a direct gaze and fearful expressions paired with an averted gaze have been interpreted in terms of a contextual evaluation of the displayed emotion during its processing (*Adams and Kleck, 2003*; *Sander et al., 2007*; *Adams et al., 2012*). In particular, 'appraisal' theories (*Sander et al., 2007*) emphasize that an angry expression paired with a direct gaze can be interpreted as behaviorally 'relevant' to the observer as being the target of a verbal or physical assault, whereas a fearful expression looking aside from the observer might signal a source of danger in the immediate vicinity of the observer. However, the mechanisms which instantiate the proposed contextual evaluation of emotions as a function of their implied threat for the observer have remained unclear. Gaze direction could either bias the perceived emotion towards its most relevant (threat-signaling) interpretation – i.e., anger when paired with direct gaze, or fear when paired with averted gaze, or increase the sensitivity to the most relevant emotion. The present study answers directly this issue by showing, both behaviorally (by comparing quantitative fits of the two effects to the behavioral data) and neurally (by regressing brain signals against emotion strength), that the improved recognition accuracy for threat-signaling emotions corresponds to a selective neural enhancement of perceptual sensitivity to these combinations of gaze and emotion.

Emotion information modulated EEG signals at centro-parietal electrodes from 500 ms following face presentation until response execution, a finding in accordance with the 'supramodal' signature of perceptual integration reported in previous studies (*O'Connell et al., 2012*; *Wyart et al., 2012a*). This centro-parietal positivity has been proposed to encode a 'domain-general' decision variable, as it varies with the strength of sensory evidence for both visual and auditory decisions, independently from the associated response (*O'Connell et al., 2012*). Here, the same centro-parietal positivity was found to increase with the emotion strength of facial expressions – which indexes the decision variable in our emotion categorization task. Importantly, the strength of this relationship was enhanced for threat-signaling emotions. This improved neural representation of threatening combinations of gaze and emotion cannot be explained by increased attentional or surprise responses, since the centro-parietal 'P3' potential, previously reported to vary as a function of attentional resources (*Johnson, 1988*) and surprise (*Mars et al., 2008*), was not increased in response to threat-signaling emotions. Moreover, we could also rule out the possibility that this enhanced neural encoding is triggered indirectly by an increase in selective attention, which should have been associated with an improved 'decoding' of participants' decisions from their underlying neural signals (*Nienborg and Cumming, 2009*, *2010*; *Wyart et al., 2015*). We therefore hypothesize that the enhanced neural processing of threat-signaling emotions proceeds in an attention-independent, bottom-up fashion.

Earlier contextual modulations of emotion processing were also observed in ventral face-selective areas from 170 ms following face presentation. While these findings contradict a 'two-stage' view according to which emotion and gaze information would be processed independently during the first hundreds of millisecond (*Pourtois et al., 2010*) before being integrated as a function of their significance to the observer (*Klucharev and Sams, 2004*), they are in agreement with recent findings (*Conty et al., 2012*; *El Zein et al., 2015*) of early interactions between emotion and gaze information on N170 and P200 components. At these early latencies, only threat-signaling emotions were encoded by face-selective neural signals, reflecting a faster processing of emotions signaling an immediate threat to the observer as a function of their associated gaze.

More strikingly, gaze direction also modulated the encoding of emotional expressions in effector-selective regions, in parallel with the effects observed in ventral face-selective areas: only threat-signaling emotions were encoded in response preparation signals overlying human motor cortex at 200 ms following face presentation. Recent work sheds light on the adaptive function of this early

representation of threat signals in motor cortex. Disrupting this motor representation using TMS impairs the facial recognition of negative (i.e., potentially threatening) emotions, not positive ones (*Balconi and Bortolotti, 2012*; *2013*). Moreover, the perception of natural scenes engages the motor cortex at very early latencies only when the emotional valence of the scene is negative (*Borgomaneri et al., 2014*). Taken together, these findings support a strong connection between emotion and motor circuits (*Grèzes et al., 2014*) enabling the brain to react swiftly and efficiently to threat signals (*Ohman and Mineka, 2001*; *Frijda, 2009*). Our findings build on these earlier observations by showing that the brain encodes parametrically the strength of threat signals in motor cortex in parallel to their representation in face-selective, sensory regions.

Finally, our data reveal a clear functional dissociation between face- and effector-selective regions as a function of individual anxiety. The enhanced sensitivity to threat-signaling emotions in face-selective temporal cortex is driven by low-anxious observers, whereas the early enhancement measured in motor cortex is only found in high-anxious observers. The observation that high-anxious individuals encode all negative emotions as equally (and strongly) salient in face-selective regions is consistent with earlier reports of a 'hyper-vigilance' to potentially threatening signals in these individuals (*Bishop, 2007*; *Cisler and Koster, 2010*), and with their tendency to interpret ambiguous stimuli as threatening (*Beck et al., 1985*) – both associated with amygdala hyperactivity (*Bishop, 2007*; *Etkin and Wager, 2007*). Nevertheless, our findings reveal that high-anxious individuals are capable of encoding threat signals in a selective fashion in motor cortex. Consistent with the idea of a compensatory mechanism, the distinct neural enhancements of temporal and motor activity found in low- and high-anxious individuals lead to similar behavioral improvements in terms of perceptual sensitivity to threat signals. Together, this pattern of findings suggests that anxiety increases the relative contribution of the motor pathway during the processing of negative social signals, in accordance with the adaptive function of anxiety in detecting efficiently and reacting swiftly to threats in the environment (*Bateson et al., 2011*). It is worth noting that the present study only involved participants with anxiety scores within the range of the healthy adult population (*Spielberger, 1983*), leaving open the question as to whether clinically anxious individuals would similarly recruit their motor cortex in response to threatening social stimuli. Moreover, further research should assess the specificity of these anxiety-dependent effects, in light of the growing evidence in favor of comorbidity between anxiety and depressive disorders.

By applying theoretical models of decision-making to socially-relevant stimuli, we were able to characterize the neural and computational mechanisms underlying the integration and interpretation of facial cues in the implicit context of threat. Evolutionary pressure might have shaped the human brain to prioritize threat signals in parallel in sensory and motor systems (*Darwin, 1872*; *LeDoux, 2012*). Such prioritization – found to proceed in a fast, selective, yet attention-independent fashion – could increase perceptual sensitivity to other features of the sensory environment (*Phelps et al., 2006*) to enable rapid and adaptive responses in complex, multidimensional situations of danger.

## Materials and methods

### Subjects

Twenty-four healthy subjects (12 females; mean age, 22.7 ± 0.7 years) participated in the EEG experiment. All participants were right-handed, with a normal vision and had no neurological or psychiatric history. They provided written informed consent according to institutional guidelines of the local research ethics committee (Declaration of Helsinki) and were paid for their participation.

### Stimuli

Stimuli consisted of 36 identities (18 females) adapted from the Radboud Faces Database (*Langner et al., 2010*) that varied in emotion (neutral, angry or fearful expressions) and gaze direction (direct toward the participant or averted 45° to the left or right). Using Adobe Photoshop CS5.1 (Adobe Systems, San Jose CA), faces were modified to remove any visible hair, resized and repositioned so that eyes, nose and mouth appeared within the same circumference. All images were converted to greyscale and cropped into a 280 x 406 pixel oval centered within a 628 x 429 pixel black rectangle.

To vary the intensity of emotional expressions, faces were morphed from neutral to angry expressions and from neutral to fearful expression using FantaMorph (Abrosoft http://www.fantamorph.com/). At first, we created 7 levels of morphs from neutral to angry expressions and from neutral to fearful expressions (separately for direct and averted gaze stimuli) using a simple linear morphing transformation. This resulted in 30 conditions for each identity: 7 levels of morphs * 2 emotions * 2 gaze directions = 28 and 2 neutral stimuli with direct and averted gaze. We then calibrated the morphing between angry and fearful expressions by performing an intensity rating pre-test of the emotional expressions and adjusting the morphs based on the results. 19 subjects (9 females, mean age, $24.7 \pm 0.9$ years) were presented with the facial expressions for 250 ms and rated the emotional intensity perceived on a continuous scale from "not at all intense" to "very intense" using a mouse device (with a maximum of 3 seconds to respond). We adjusted for differences between emotions by linearizing the mean curves of judged intensities and creating corresponding morphs that were validated on 10 new subjects (4 females, mean age $24.1 \pm 1.9$).To summarize, the stimuli comprise of 36 identities with an Averted gaze condition and a Direct gaze condition, each with 7 levels of Anger and 7 levels of Fear equalized in perceived emotional intensities and a neutral condition, resulting in a total of 1080 items (see *Figure 2a* for examples of stimuli).

## Experimental procedure

Using the Psychophysics-3 Toolbox (*Brainard, 1997*; *Pelli, 1997*), stimuli were projected on a black screen. Each trial was initiated with a white oval delimiting the faces that was kept during all the trial. The white oval appeared for approximately 500 ms, followed by a white fixation point presented at the level of the eyes for approximately 1000 ms (to keep the fixation to the upcoming faces natural and avoid eye movements from the center of the oval to eye regions), than the stimuli appeared for 250 ms. Participants' task was to decide whether the faces expressed Anger or Fear by pressing one of the two buttons localized on two external devices held in their right and left hands, with their right or left index correspondingly (*Figure 2b*). An Anger/Fear mapping was used (e.g Anger: Left hand, Fear: Right hand) kept constant for each subject, counterbalanced over all subjects. All stimuli were presented once, resulting in a total of 1080 trials. The experiment was divided in 9 experimental blocks, each consisting of 120 trials, balanced in the number of emotions, directions of gaze, gender and levels of morphs. After each block, the percentage of correct responses was shown to the participants to keep them motivated.

## Behavioral data analyses

Repeated-measures ANOVA was performed on the percentage of correct responses and average reaction times, with gaze direction (direct/averted), emotion (anger/fear), and intensity (7 levels of morphs) as within-subjects factors.

## Model selection

We performed model-guided analyses of the behavioural data to characterize the observed increase in recognition accuracy for THREAT+ combinations of gaze and emotion. We used Bayesian model selection based on the model evidence (estimated by a 10-fold cross-validation estimation of model log-likelihood, which penalizes implicitly for model complexity without relying on particular approximations such as the Bayesian Information Criterion or the Akaike Information Criterion). We applied both fixed-effects and random-effects statistics previously described in the literature. The fixed-effects comparison assumes all participants to have used the same underlying model to generate their behavior, such that the overall model evidence for a given model is proportional to the product of model evidence for the model for all participants. Based on this model evidence, we compared different models by computing their Bayes factor as the ratio of model evidence of the compared model (*Jeffreys, 1961*; *Kass and Raftery, 1995*). The random-effects comparison is more conservative in allowing different participants to use different models to generate their behavior, and aims at inferring the distribution over models that participants draw from (*Penny et al., 2010*). For this comparison, we computed support for the winning model by the exceedance probability ($p_{exc}$), which is the probability that participants were more likely to choose this model to generate behavior over any alternative model.

We started with the simplest model (model 0) that could account for each subject's decisions using a noisy, 'signal detection'-like psychometric model to which we included a lapse rate, thereby considering that subjects guessed randomly on a certain proportion of trials:

$$P(anger) = \phi[w*x+b]*(1-\varepsilon)+0.5*\varepsilon$$

where P(anger) corresponds to the probability of judging the face as angry, Φ[.] to the cumulative normal function, w to the perceptual sensitivity to the displayed emotion, **x** to a trial-wise array of evidence values in favor of anger or fear (emotion strength, from −7 for an intense expression of fear to +7 for an intense expression of anger), b to an additive, stimulus-independent bias toward one of the two emotions, and ε to the proportion of random guesses among choices. We compared a 'null' model which did not allow for contextual influences of gaze direction on the decision process, to two additional models which instantiate two different mechanisms which could account for the observed increase in recognition accuracy for THREAT+ combinations of gaze and emotion. A first possibility (model 1) would be that gaze direction biases emotion recognition in favor of the interpretation signaling higher threat (anger for a direct gaze, fear for an averted gaze). Alternatively (model 2), gaze direction might selectively increase sensitivity to emotions signaling higher threat in this context (modeled by a different sensitivity to emotions in THREAT+ vs. THREAT− conditions).

## EEG acquisition and pre-processing

An EEG cap of 63 sintered Ag/AgCl ring electrodes (Easycap) was used to record EEG activity. EEG activity was recorded at a sampling rate of 1000 Hz using a BRAINAMP amplifier (Brain Products, BRAINAMP MR PLUS) and low pass filtered online at 250 Hz. The reference channel was placed on their nose and a forehead ground was used. Impedances were kept under a threshold of 10 kΩ.

The raw EEG data was recalculated to average reference, down-sampled to 500 Hz, low-pass filtered at 32 Hz, and epoched from 1 s before to 4 s after the face stimulus onset using EEGLAB (*Delorme and Makeig, 2004*). First, EEG epoched data was visually inspected to remove muscle artifacts and to identify noisy electrodes that were interpolated to the average of adjacent electrodes. Second, independent component analysis (ICA) that excluded interpolated electrodes was performed on the epoched data and ICA components capturing eye blink artifacts were manually rejected. A last, visual inspection was done on the resulting single epochs to exclude any remaining trials with artifacts. After trial rejections, an average of 999 ± 10 trials per subject remained.

Time frequency analysis was performed using the Fieldtrip toolbox for MATLAB (*Oostenveld et al., 2011*). We were particularly interested in motor mu-bands (8–32 Hz) and thus estimated the spectral power of mu-beta band EEG oscillations using 'multitapering' time frequency transform (Slepian tapers, frequency range 8–32 Hz, five cycles, three tapers per window). The purpose of this multitapering approach is to obtain more precise power estimates by smoothing across frequencies. Note that this time–frequency transform uses a constant number of cycles per window across frequencies, hence a time window whose duration decreases inversely with increasing frequency.

## EEG analyses

### Time frequency: motor lateralization measures

As the suppression of mu-beta activity in the hemisphere contralateral to the hand used for response is a marker of motor preparation to response (*Donner et al., 2009*; *de Lange et al., 2013*), spectral power from 8 to 32 Hz were calculated at each electrode and time point for all subjects. Then for each subject, to obtain the lateralization measures, the spectral power from 8 to 32 Hz for the trials where the subjects responded with their right hand was subtracted from that of the trials where the subjects responded with their left hand. After averaging on all subjects, electrodes where the motor lateralization was maximal from 200 ms before to response time were selected: 'P3','CP3','C3' for the left hemisphere and 'P4','CP4','C4' for the right hemisphere. Motor lateralization specific to 'anger' or 'fear' responses was obtained by taking into account the Anger/Fear mapping used and subtracting 'Anger' hand spectral activity to 'Fear' hand spectral activity (the average on 'P3','CP3','C3' minus the average on 'P4','CP4','C4' if participants responded 'Anger' with the left hand and vis versa if they responded 'Anger' with the right hand).

## Regression analysis: encoding of the emotional information

In our emotion categorization task, evidence strength corresponds to the intensity of the displayed emotion. On the basis of recent studies (*Wyart et al., 2012b*, *2015*), we therefore performed single-trial regressions of EEG signals against this variable. A general linear regression model (GLM) was used where emotion strength (from 0 for a neutral/emotionless expression to 7 for an intense fear/anger expression) was introduced as a trial-per-trial predictor of broadband EEG signals at each time point after stimulus onset (from 200 ms before to 1 s after stimulus onset), at each electrode. The corresponding parameter estimates of the regression, reported in arbitrary units, were measured per participant, and then averaged across participants to produced group-level averages. The time course of the parameter estimates describes the neural 'encoding' of the relevant (emotion) information provided by the presented facial expression. Electrodes and time points where the parameter estimates of the regression were maximal were selected to further compare between the conditions of interest: Anger Direct and Fear Averted vs Fear Direct and Anger Averted.

Similar general linear regressions were also performed on lateralized mu-beta activity. Once more, the intensity of the emotional expression was entered as a regressor to predict the trial-per-trial motor lateralization activity (calculated as described above) for each time point after stimulus onset. The only important difference is that owing to the 'signed' nature of the motor lateralization index (positive for a contra-lateralized activity), we expressed the intensity of the emotional expression as signed by the displayed emotion, from -7 for an intense expression of fear to +7 for an intense expression of anger.

## Neural-choice correlation analyses

We determined whether residual fluctuations in single-trial EEG signals unexplained by variations in emotion strength (measured by the previous neural regressions against emotion strength) modulated the recognition of the subsequent emotion. This approach is reminiscent of 'choice probability' measures applied in electrophysiology to measure correlations between neural activity and choice behavior (*Britten et al., 1996*; *Shadlen et al., 1996*; *Parker and Newsome, 1998*) – by estimating how much fluctuations in recorded neural signals are 'read out' by the subsequent decision (*Wyart et al., 2012a*, *2015*). The advantage of measuring neural-choice correlations within the framework of our computational model is that we could not only establish *whether*, but also *how* neural fluctuations influenced the subsequent behavior – either additively as a stimulus-independent bias, or multiplicatively as a change in perceptual sensitivity.

In practice, we estimated the parameters $b_{mod}$ and $w_{mod}$ of these neural modulation terms at each time point following face presentation via an EEG-informed regression of choice for which the neural residuals $e$ from the regression against emotion strength were entered either alone (additive influence, parameter $b_{mod}$, model 1) or as their interaction with the strength of the displayed emotion (multiplicative influence, parameter $w_{mod}$, model 2) as an additional predictor of the subsequent categorical choice, as follows:

$$1.\ p(anger) = \phi(w \cdot x + b + b_{mod} \cdot e)$$

$$2.\ p(anger) = \phi(w \cdot x + b + w_{mod} \cdot e \cdot x)$$

We applied Bayesian model selection to compare between these two possible modulations of the decision process by neural fluctuations using both fixed-effects and random-effects statistical procedures described above.

## EEG statistical procedures

All regression-based analyses of the EEG data were performed independently for each subject, and then followed by a second-level analysis at the group level to assess the significance of the observed effects across participants. Second-level analyses relied on standard parametric tests (t-tests, repeated-measures analyses of variance), with explicit control over the type-1 error rate arising from multiple comparisons across time points through non-parametric cluster-level statistics as described in (*Maris and Oostenveld, 2007*). The pairing between experimental conditions and EEG signals was shuffled pseudo-randomly 1,000 times, and the maximal cluster-level statistics (the sum of t-values across contiguously significant time points at a threshold level of 0.05) were extracted for each shuffle to compute a 'null' distribution of effect size across a time window of [-200,+1000] ms around

stimulus presentation, or [-1000,+200] around response onset. For each significant cluster in the original (non-shuffled) data, we computed the proportion of clusters in the null distribution whose statistics exceeded the one obtained for the cluster in question, corresponding to its 'cluster-corrected' p-value. We applied a second bootstrapping method to test for significant shifts in neural encoding latencies between conditions, using the 'jackknifing' procedure described in (*Kiesel et al., 2008*).

Bayes factors were computed for critical absence of effects observed, to distinguish between the lack of sensitivity of tests and genuine absence of difference (*Dienes, 2011*). A group-level random-effects Bayes factor was computed under the same assumptions as a standard *t* test that states that the distribution of the observed effect across individuals can be approximated by a normal distribution of the mean (μ) and standard deviation (σ). We computed the maximum log-likelihood of the model in favor of the "null" hypothesis, which assumes that μ=0 and the model in favor of the "effect" hypothesis, for which both μ and σ can be adjusted freely to the observed data. We then used the Bayesian information criterion to compare the two models and compute the corresponding Bayes factor. A Bayes factor below 1/3 provides substantial evidence in favor of the null hypothesis whereas a Bayes factor > 3 provides in favor of the effect hypothesis.

We performed control analyses to confirm the robustness of the anxiety-threat correlation across individuals observed in temporal and motor regions. For this purpose, we performed a leave-one-out, cross-validation procedure in which we computed 'cross-validated' prediction intervals for the group-level regression line at the anxiety score of participant #n when the data for participant #n was excluded from the group-level regression. This procedure was repeated for the 24 participants for the anxiety-threat correlation in motor areas and showed that the neural effects of two participants fell slightly (< 20%) outside of the cross-validated 95% prediction intervals. Recomputing the group-level correlation in motor areas after excluding the two outliers, leaving 22 participants in the analysis, led to a significant effect (*r* = 0.48, d.f. = 20, p=0.02). This leave-one-out analysis identified no outlier for the anxiety-threat correlation in temporal regions.

## Source reconstruction analysis

Source analysis was performed using Brainstorm (*Tadel et al., 2011*). A source model consisting of 15,002 current dipoles was used to calculate Kernel inversion matrices for each subject based on all the trials of the subject. Dipole orientations were constrained to the cortical mantle of a generic brain model taken from the standard Montreal Neurological institute (MNI) template brain provided in brainstorm. Individual scalp models, recorded with a Zebris device, were used to warp this template head model to EEG sensor caps. Using the OpenMEEG BEM model (*Kybic et al., 2005*; *Gramfort et al., 2010*), the forward EEG model was computed for each subject. Individual inversion matrices (15002 vertices * 63 electrodes) were then extracted to perform single trial regressions at the source level.

## Threat and trustworthiness rating experiment

20 subjects participated to the experiment (10 females, mean age = 22.7 ± 0.6). The 36 identities used in the experiment were presented in the neutral condition only. Each identity was presented twice, once with a direct, and once with an averted gaze. Faces appeared on the screen for 2 seconds after which they disappeared and 2 continuous scales were drawn on the screen. Participants rated the identities on these scales in terms of threat and trustworthiness from "not at all" to "very much" (a text appeared at the top of the scales reminding the instructions: How much is this face threatening/trustworthy?). The order of the scales was randomized across subjects. The scales stayed on the screen until the two responses were given, however subjects were instructed to answer intuitively without spending too much time to decide.

## Acknowledgements

This work was supported by grants from the French National Research Agency ANR-11-EMCO-00902, ANR-11-0001-02 PSL*, ANR-10-LABX-0087, the Fondation ROGER DE SPOELBERCH and by INSERM. We wish to thank Laurent Hugueville for his useful technical help.

# Additional information

## Funding

| Funder | Grant reference number | Author |
|---|---|---|
| Agence Nationale de la Recherche | ANR-11-EMCO-00902 | Julie Grèzes |
| Agence Nationale de la Recherche | ANR-11-0001-02 PSL* | Valentin Wyart Julie Grèzes |
| Agence Nationale de la Recherche | ANR-10-LABX-0087 | Valentin Wyart Julie Grèzes |
| Fondation Roger de Spoelberch | | Julie Grèzes |
| Institut national de la santé et de la recherche médicale | | Valentin Wyart Julie Grèzes |

The funders had no role in study design, data collection and interpretation, or the decision to submit the work for publication.

## Author contributions

MEZ, Conception and design, Acquisition of data, Analysis and interpretation of data, Drafting or revising the article; VW, JG, Conception and design, Analysis and interpretation of data, Drafting or revising the article

## Author ORCIDs

Valentin Wyart, http://orcid.org/0000-0001-6522-7837

## Ethics

Human subjects: All participants provided written informed consent according to institutional guidelines of the local research ethics committee (following the Declaration of Helsinki) and were paid for their participation. In this consent, they agreed that their data would be stored anonymously, analyzed, and that the corresponding results would be presented in conferences and published in peer-reviewed journals. Ethical approval was provided by the local ethics committee (Comite de Protection des Personnes, Ile-de-France VI, Inserm approval #C07-28, DGS approval #2007-0569, IDRCB approval #2007-A01125-48), in accordance with article L1121-1 of the French Public Health Code.

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
