## [Decision Letter]

Thank you for submitting your work entitled "Anxiety dissociates the adaptive functions of sensory and motor response enhancements to social threats" for peer review at *eLife*. Your submission has been favorably evaluated by Timothy Behrens (Senior editor) and three reviewers, one of whom, Peggy Mason, is a member of our Board of Reviewing Editors.

The reviewers have discussed the reviews with one another and the Reviewing editor has drafted this decision to help you prepare a revised submission.

The interaction of emotional expression with gaze direction dependent on gain sensitivity is elegant and well done, and novel. The anxiety dependent analysis and related findings are intriguing. Addressing the following issues would make this a valuable contribution.

Most importantly, the Methods should be incorporated into the Results. The Methods are too integral to the understanding of the findings to be presented separately. Please write the manuscript to be understandable to the general reader. Clarity is more important than standard format and all reviewers agreed that methodological details should be interwoven into the Results presentation.

Can the authors justify not considering participant response in their EEG analysis? Or possibly analyzing the data in the standard way – by stimulus – and then additionally by classification response?

There is concern that one outlier is driving the anxiety-threat correlation for motor areas. Could the authors please address this concern?

Please explicitly state how multiple comparisons were accounted for.

Were pts screened for social anxiety or depression?

The above points are the major ones that drove the reviewers' decisions. Complete reviews are presented below.

*Reviewer #1:*

This is an interesting study with an even more interesting answer once the anxiety bit is thrown in. I don't have many comments in part because the EEG methods are opaque to me. My one major concern is whether the middle portion of the Results – concerning all subjects together – has value. It appears that the Results are in fact driven by two different strongly significant results in two different subject cohorts – high and low anxiety. So the overall increased sensitivity to threat+ is constructed of a low anxiety - perception and high anxiety = motor pieces.

*Reviewer #2:*

In general I think this is a nice piece of work and the authors have clearly gone to considerable effort to provide a full account of their data set and are to be applauded for the extent of the analysis conducted to better understand the neural response to threatening social stimuli. However, I found some of the analysis difficult to parse and in some cases important information was missing (or not obvious).

Particular points that bothered me were the selective presentation of the behavioural results. Where are the reaction time results (as mentioned to have been analysed in the Methods)? Was there no interaction between the categorisation performance and level of morph when exploring the eye gaze-emotion performance interaction? The plots in Figure 2 suggest that as one might expect, the effect of eye gaze is stronger for the weaker emotion signals (closer to neutral). As a more minor point, why are participants not 100% correct for the end points of the morphs (100% anger, 100% fear) – are some identities consistently categorised incorrectly?

Turning to the EEG analysis. The authors employ advanced and relatively specialised methods to explore the relationship of the EEG signal to the modulations in emotional strength and their results are interesting and appear strong. While I think they have done a reasonable job in explaining these methods and how they should be interpreted, I would have liked more detail on the statistical tests they employed. Surely they performed some bootstrapping to correct for the multiple comparisons issues they face. This is hinted at in the caption to Figure 3 (cluster corrected threshold) but no detail is provided anywhere that I can see.

Furthermore, I think the authors used all trials (correct and incorrect) in their EEG analysis but it’s unclear to me why this would be justified, particularly for the weaker emotional stimuli (where participants are only correct about 70% of the time in categorising them). For example a threat+ fear averted trial could have been categorised by the participant as an anger face (and thus treated by the participant as threat-). Would the authors expect the neural encoding to be the same in both cases? If not, how can they justify not considering participant response in their EEG analysis?

Finally I enjoyed the anxiety dependent analysis and the findings are clear and interesting.

*Reviewer #3:*

Wyart et al. examine the interaction of emotional expression and gaze direction and test two models: multiplicative threat enhancement of emotion perception, versus additive response bias. The two models are explained clearly with reference to the behavioural data, and thereafter the authors test these models against single trial neural data. The authors report the time course of sensitivity to high versus low threat conditions. They also test bias versus sensitivity models, using the residuals from the regression analysis, at both temporal partial electrodes, and find an enhancement in perceptual sensitivity and also against test the models against a motor lateralization index, and find a bias effect.

This part of the paper is elegant and well done, and to my knowledge, novel. The Methods section is really clear – my preference would be to see some of the explanation of the models moved into the main part of the paper, as this part, while 'technical', is really needed to understand the meaning of the subsequent analysis, and as it stands, the introduction is much less clear than the longer explanation.

The final section of the paper claims a relationship with anxiety levels in individual subjects, as threat+ and - are differently modulate temporal areas in low anxious individuals), and threat+ modulates motor areas in high anxious individuals.

I have reservations about the second part of the dissociation in high anxious individuals mostly to do with Figure 7 – it seems to me to be the main weakness.

1) Anxiety is co-morbid with depression, and social anxiety. Were participants screened for their scores with other factors that might be correlated with anxiety?

2) All participants bar one lie in the very centre of the normal range. In Figure 7, it looks like that 40+ score might be an outlier. R values are computed with Pearson (I assume, though I notice that isn't on the figure; it should be). If the outlying subject is excluded, does the correlation survive?

Following on from that, it seems to me that there is an increase in variabilityin the 'more anxious' group, which could be driving the correlation, especially as it looks as though the difference between threat and no-threat in the high anxious group in 7B isn't significant.

3) The effect in Figure 7 seems to be a weak effect at a single time point. There is no justification/explanation of the selection of either the window in 7A, or the single point in 7B (if that is what it is).

4) The role of reaction time: the authors say that they found no difference in the behavioural responses between the more and less anxious groups. Did that include reaction times?

[Editors' note: further revisions were requested prior to acceptance, as described below.]

Thank you for resubmitting your work entitled "Anxiety dissociates the adaptive functions of sensory and motor response enhancements to social threats" for peer review at *eLife*. Your submission has been favorably evaluated by Timothy Behrens (Senior editor) and Peggy Mason (Reviewing editor).

This is an excellent revision. You have utilized the reviews to make this complex paper as understandable as its interesting findings deserve. The editor has three very minor suggestions for clarifications (1-2 sentences per issue should do it). The issues are:

Figure 1 is very useful *except* I didn't understand the emotion axis in Figure 1 (filled in gray area) until I saw the bottom of Figure 3 where it is labeled as emotion sensitivity. Please label in Figure 1 and explain in the legend (in addition to the explanation in the text).

How were the times in Figure 4 chosen? The times in B and C make some sense to me but I don't understand how 280 was chosen when it is so far from the region of significant differences. Also please explain why the maps in D-F don't look as though they came from the brains mapped in the middle of A-C (particularly B-C). Upon re-reading this, I think I understand the timing issue – you are mapping the deviations from 0 rather than the peak differences between the conditions. Perhaps make the emphasis on the latter equal to that on the former by making the gray boxes into bars up top to match the green and orange bars? All equally emphasized that way. Confusion regarding map alignments or lack thereof remains.

For the general reader, could you add a word of background about the lateralization difference illustrated in Figure 6 and discussed in the subsection “Early neural encoding of threat-signaling emotions in motor preparation”? It appears to be a robust difference – is it known and expected? Surprising in any way? Remember that the *eLife* readership is broad and not necessarily in the know about much intra-field background.

---

## [Author Response]

The interaction of emotional expression with gaze direction dependent on gain sensitivity is elegant and well done, and novel. The anxiety dependent analysis and related findings are intriguing. Addressing the following issues would make this a valuable contribution.

*Most importantly, the Methods should be incorporated into the Results. The Methods are too integral to the understanding of the findings to be presented separately. Please write the manuscript to be understandable to the general reader. Clarity is more important than standard format and all reviewers agreed that methodological details should be interwoven into the Results presentation.*

We followed the reviewers’ advice by blending the description of the methods into the main text. We aimed to make the findings understandable to non-specialists without requiring them to read in length the Methods section. In doing so, we chose to keep methodological details in the Methods section while providing the rationale of regression-based analyses in the main text. We believe that these changes significantly improved the readability of the manuscript to non-specialists, and we thank the reviewers for this useful suggestion.

*Can the authors justify not considering participant response in their EEG analysis? Or possibly analyzing the data in the standard way – by stimulus – and then additionally by classification response?*

We believe this comment to be tightly linked to the previous point – describing the methods in the main text should make it clear why we did not perform straightforward contrasts based on participants’ responses (e.g., correct responses vs. errors). We did not perform such contrasts because the corresponding effects can be mediated indirectly (and trivially) by changes in the stimulus itself – not in the decision process. For example, both EEG signals and the accuracy of behavioral responses grow with the strength of the emotion displayed by the face stimulus, such that classifying EEG signals with respect to response accuracy could show a significant difference which is merely due to the fact that correct responses were on average associated with stronger emotions – i.e., by external differences in stimulation. Performing a response classification analysis not confounded with changes in stimulation would have required a ‘constant stimulus’ condition, e.g., considering emotionless/neutral stimuli in isolation (only 1/15 of all trials in our experimental design). The other approach, which we have followed throughout the manuscript and which allows to analyze all trials, is the regression-based method consisting in studying how trial-by-trial *residuals* from the regression of stimulus features (here, emotion strength) against EEG signals co-vary with response accuracy. The other advantage of model-based regression analyses, which do take participants’ responses into account and which we underline more explicitly in the revised manuscript, is that they allow not only to assess whether EEG signals co-vary with behavioral responses, but also how they do so: through modulations of perceptual sensitivity and/or bias.

On the theoretical side, it is worth noting that according to decision theories inspired by classical Signal Detection Theory, the decision process underlying correct responses and errors is the same – the only difference lies in the position of the noisy decision variable with respect to the decision criterion on particular trials. Identifying sensitivity and bias is therefore informed by both correct responses and errors. Following this reasoning, we chose to perform regression-based analyses of the EEG data without artificially separating correct responses from errors. The resulting pattern therefore indicates the enhancement of neural sensitivity to threat-signaling emotions *irrespective of the accuracy of the subsequent response*. Nevertheless, even when considering only correct trials in the ‘encoding’ analyses shown in Figure 4, significant differences between THREAT+ and THREAT- emotions emerge at temporal electrodes at 170 ms following stimulus onset (*t*_23_= -2.1, *p* < 0.05), parietal electrodes at 500 ms (*t*_23_= 4.2, *p* < 0.001) and motor signals at 200 ms (*t*_23_= 3, *p* < 0.01). We added these results in the Results section:

“This threat-dependent enhancement remained significant when considering only correct responses (temporal: *t*_23_ = -2.1, *p* < 0.05; centro-parietal *t*_23_ = 4.2, *p* < 0.001).”,

and:

“This early threat-dependent motor enhancement remained significant when considering only correct responses (*t*_23_ = 3.0, *p* < 0.01).”

We hope that these clarifications, which are mirrored by additional descriptions of the regression-based methods throughout the text, will make the main findings more easily understandable to the broad readership of *eLife*.

*There is concern that one outlier is driving the anxiety-threat correlation for motor areas. Could the authors please address this concern?*

We performed additional control analyses to rule out the possibility that the anxiety-threat correlation observed in motor regions is driven by one or few outliers. First, it is important to note that the anxiety scores of our participants (between 20 and 45) fall well within the range of the normal population (< 60), and there is thus no a priori reason to exclude any of our participants solely based on his/her anxiety score. Besides, it is precisely the between-participant variability in anxiety scores that afforded their correlation with neural effects. As a first control analysis, we computed prediction intervals for the group-level regression line at the anxiety score of each participant. No participant fell outside of the 95% prediction intervals for both temporal (Figure 8) and motor areas (Figure 8) – and could thus be labeled statistically as outlier.

We performed a second, stricter control analysis to further confirm the robustness of the anxiety-threat correlation observed in temporal and motor regions. We performed a leave-one-out, cross-validation procedure in which we computed ‘cross-validated’ prediction intervals for the group-level regression line at the anxiety score of participant #n when the data for participant #n was excluded from the group-level regression. This procedure was repeated for the 24 participants for the anxiety-threat correlation in motor areas and showed that the neural effects of two participants fell slightly (< 20%) outside of the *cross-validated* 95% prediction intervals – including participant #4 whose anxiety score was 45. Nevertheless, recomputing the group-level correlation in motor areas after excluding the two outliers, leaving 22 participants in the analysis, led to a significant effect (*r* = 0.48, d.f. = 20, *p* = 0.02). This leave-one-out analysis identified no outlier for the anxiety-threat correlation in temporal regions.

Together, we believe that these control analyses confirm the robustness of the anxiety-threat correlations for both temporal and motor regions, and we now describe them in the Methods section of the revised manuscript.

*Please explicitly state how multiple comparisons were accounted for.*

We apologize for not providing enough statistical details about correction for multiple comparisons in the Methods section. We applied conventional corrections described in (Maris and Oostenveld, 2006), reflected in the ‘cluster-corrected’ p-values reported in the Results section. We now explicitly state how multiple corrections were accounted for in the Methods section of the revised manuscript in a new ‘statistical procedures’ subsection (“All regression-based analyses […] corresponding to its ‘cluster-corrected’ p-value.”).

*Were pts screened for social anxiety or depression?*

This question is related to the growing evidence in favor of comorbidity between anxiety (related to six anxiety disorders among which social anxiety) and depressive disorders, which calls for the identification of both specific and shared neurocognitive markers of anxiety and depression.

Our findings put forward the adaptive and *positive* function of moderate (non-clinical) anxiety in response to social threats (Bateson et al., 2011). Our participants filled the STAI questionnaire, the most commonly used in experimental investigations of anxious features in non-clinical samples (see, e.g., Sylvers et al., 2011), but they were neither screened for depression nor for social anxiety. As stressed recently by Grupe and Nitschke (2013): “Despite its lack of specificity, the relevance of research using STAI is underscored by its sensitivity as a marker of risk for anxiety disorders.” Furthermore, concerning the comorbidity between depression and anxiety, it has been shown that patients with clinically defined anxiety and depression had lower mortality rates than those with depression alone (Mykletun et al., 2009). This finding thus suggests that both non-clinical and clinical anxiety may have a specific ‘protective’ function, related to the adaptive role of anxiety in survival, which is not shared with depression.

We nevertheless agree with the reviewers and acknowledge the resulting limitation as to the specificity of the observed anxiety-dependent effects. Consequently, we have added a sentence to the Discussion section:

“Further research should assess the specificity of these anxiety-dependent effects, in light of the growing evidence in favor of comorbidity between anxiety and depressive disorders.”

Reviewer #1:

*This is an interesting study with an even more interesting answer once the anxiety bit is thrown in. I don't have many comments in part because the EEG methods are opaque to me. My one major concern is whether the middle portion of the Results – concerning all subjects together – has value. It appears that the Results are in fact driven by two different strongly significant results in two different subject cohorts – high and low anxiety. So the overall increased sensitivity to* threat*+ is constructed of a low anxiety - perception and high anxiety = motor pieces.*

We thank the reviewer for his/her positive comments. We agree with the reviewer that the sensory and motor response enhancements are strongly modulated, in a doubly dissociable fashion, by the anxiety score of the participants. Nevertheless, we believe the middle portion of the Results section to be of importance for the field, as it sheds light on the specific neural process responsible for threat-dependent enhancements: an increased neural sensitivity to threat-signaling emotions rather than a perceptual or decision bias, which confirms our model-based analysis of the behavioral data and which had not been tested before to our knowledge. The later dissociation between high- and low-anxious participants builds upon these first findings by highlighting the selective contributions of sensory and motor regions to the increased sensitivity to threat-signaling emotions in the two high- and low-anxious groups.

To make our EEG analyses clearer, and as indicated in the collective responses above, we followed the reviewer’s advice by blending the description of the methods into the main text. We aimed to make the findings understandable to non-specialists without requiring reading in length the Methods section. We hope that the reviewer will agree that these changes significantly improved the readability of the manuscript to non-specialists.

Reviewer #2:

*In general I think this is a nice piece of work and the authors have clearly gone to considerable effort to provide a full account of their data set and are to be applauded for the extent of the analysis conducted to better understand the neural response to threatening social stimuli. However, I found some of the analysis difficult to parse and in some cases important information was missing (or not obvious). Particular points that bothered me were the selective presentation of the behavioural results.*

We thank the reviewer for his/her positive evaluation of our analytic framework and findings. We understand the reviewer’s concern regarding our selective description of the behavioral results. We originally decided to favor concision over exhaustiveness in the behavioral results to facilitate the transmission of the main message, but retrospectively we can see how the missing details can impair the readability of this section. We have followed the reviewer’s suggestion by providing the missing information on recognition accuracy and reaction times in the revised manuscript.

*Where are the reaction time results (as mentioned to have been analysed in the Methods)?*

The reaction times results have been added to the Results section:

“Reaction time (RT) analyses revealed a decrease of correct RTs with emotion strength (repeated-measures ANOVA, F_6,138_ = 54.5, p < 0.001), faster responses to angry as compared fearful faces (F_1,23_ = 12, p < 0.01), and faster responses to direct as compared to averted gaze (F_1,23_ = 7.7, p < 0.05). Furthermore, an emotion by gaze interaction was observed (F_1,23_ = 8, p < 0.01), corresponding to faster reaction times for direct as compared to averted gaze in the anger condition only (t_23_ = -3.9, p < 0.001).”

Note that these reaction time results provide an independent replication of our main finding of an interaction between gaze direction and emotion on the accuracy of emotion recognition – by showing the same pattern on the speed of emotion recognition.

*Was there no interaction between the categorisation performance and level of morph when exploring the eye gaze-emotion performance interaction? The plots in Figure 2 suggest that as one might expect, the effect of eye gaze is stronger for the weaker emotion signals (closer to neutral).*

There is indeed an interaction between threat+/− and the level of morph on categorization performance, and these results have been added to the Results section:

“Moreover, a significant emotion by gaze by emotion strength interaction was observed (F_6,138_ = 4.3, p < 0.01), explained by a stronger influence of gaze on emotion categorization at weak emotion strengths (gaze by emotion interaction for levels 1 to 4, F_1,23_ = 23.8, p < 0.001) than at high emotion strengths (gaze by emotion interaction for levels 5 to 7, F_1,23_ = 5.1, p < 0.05).”

Nevertheless, note that this interaction alone is not specific of either of our two models (bias vs. sensitivity difference between threat+ and threat- conditions). The difference between the two models lies in whether the effect of gaze is strongest for emotionless/neutral stimuli (bias effect) or for weak emotion strengths (sensitivity effect, see the new Figure 1 showing model predictions for both types of effects). This observation was one of the main reasons for model-based analyses, which can arbitrate between these two alternative accounts of this interaction on emotion categorization.

*As a more minor point, why are participants not 100% correct for the end points of the morphs (100% anger, 100% fear) – are some identities consistently categorised incorrectly?*

This is a very good point, and unfortunately we had omitted to explain how we accounted for this observation in our modeling of participants’ behavior. In every model we fitted to participants’ choices, we have added an additional ‘lapse’ parameter that corresponds to the proportion of random guesses among choices. This parameter is conventionally used in psychophysics to model the imperfect asymptotic performance noted by the reviewer. We have corrected the Methods section to describe the full model including the ‘lapse’ parameter:

“We started with the simplest model (model 0) that could account for each subject’s decisions using a noisy, ‘signal detection’-like psychometric model to which we included a lapse rate, thereby considering that subjects guessed randomly on a certain proportion of trials:

P(anger) = Ф[w***x** + b]*(1− ɛ)+ 0.5*ɛ

where P(anger) corresponds to the probability of judging the face as angry, Ф[.] to the cumulative normal function, w to the perceptual sensitivity to the displayed emotion, **x** to a trial-wise array of evidence values in favor of anger or fear (emotion strength, from −7 for an intense expression of fear to +7 for an intense expression of anger), b to an additive, stimulus-independent bias toward one of the two emotions, and ε to the proportion of random guesses among choices.”

Note that the group-level mean for the lapse rate ɛ fitted simultaneously with the other parameters w and b is equal to 16.2 ± 1.8% (mean ± s.e.m.), a value that importantly did not differ between threat+ and threat- conditions (*t*_23_ = 0.4, *p* > 0.5).

Nevertheless, we took the reviewer’s alternative hypothesis seriously and performed additional analyses to rule out the possibility that some identities were consistently categorized incorrectly by the participants. First, we verified that all of the identities used in the study were categorized significantly above chance at both extremes of the emotion morph axis. This was indeed the case (paired t-tests, all *t*_23_ > 6.4, all *p* < 0.001). Second, we fitted separate sigmoid functions to describe participants’ average psychometric response curve for each identity taken in isolation. This ‘fixed-effects’ analysis confirmed that all of the identities were consistently assigned to anger and fear as a function of morph level, as indicated by significantly positive slopes of the associated psychometric functions (logistic regressions, all *t*-values > 12.8, all *p* < 0.001). Furthermore, the distribution of the point of subjective ‘neutrality’ (at which the psychometric function crosses 0.5) across identities is centered on 0.01 (on a scale going from −1 for the strongest fear expression to +1 for the strongest anger expression), with a standard deviation of 0.20, indicating that there were no extreme biases in categorization across participants for the identities used in the study. Together, we believe that these additional results fully support our choice to model the imperfect asymptotic performance by stimulus-independent lapses rather than by misrecognized emotions displayed by certain identities.

*Turning to the EEG analysis. The authors employ advanced and relatively specialised methods to explore the relationship of the EEG signal to the modulations in emotional strength and their results are interesting and appear strong. While I think they have done a reasonable job in explaining these methods and how they should be interpreted, I would have liked more detail on the statistical tests they employed. Surely they performed some bootstrapping to correct for the multiple comparisons issues they face. This is hinted at in the caption to Figure 3 (cluster corrected threshold) but no detail is provided anywhere that I can see.*

As indicated in our response to collective points above, we have added a ‘statistical procedures’ paragraph in the Methods section, where we describe in more detail the statistical analyses performed on the data. Concerning the issue of multiple comparisons, we applied conventional corrections described in (Maris and Oostenveld, 2006), reflected in the ‘cluster-corrected’ p-values reported in the Results section. We now explicitly state how multiple corrections were accounted for in the Methods section of the revised manuscript in a new ‘statistical procedures’ subsection (“All regression-based analyses […] corresponding to its ‘cluster-corrected’ p-value.”).

*Furthermore, I think the authors used all trials (correct and incorrect) in their EEG analysis but it’s unclear to me why this would be justified, particularly for the weaker emotional stimuli (where participants are only correct about 70% of the time in categorising them). For example a* threat+ *fear averted trial could have been categorised by the participant as an anger face (and thus treated by the participant as* threat-*). Would the authors expect the neural encoding to be the same in both cases? If not, how can they justify not considering participant response in their EEG analysis?*

As argued in our response to collective points above, it is important to remember that according to decision theories inspired by classical Signal Detection Theory, the decision process underlying correct responses and errors is the same – the only difference lies in the position of the noisy decision variable with respect to the decision criterion on particular trials. Identifying sensitivity and bias is therefore informed by both correct responses and errors. Following this reasoning, we decided to perform regression-based analyses of the EEG data without artificially separating correct responses from errors. The resulting pattern thus reflects the enhancement of neural sensitivity to threat-signaling emotions, irrespective of the accuracy of the subsequent response.

Nevertheless, we do not mean that the neural encoding of emotion strength should be the same when analyzing separately correct responses and errors. And as expected, the neural encoding of emotion strength by EEG signals at temporal, centro-parietal and motor electrodes did not reveal any significant cluster when considering errors in isolation (cluster-corrected *p* > 0.1). By contrast, considering only correct trials in the encoding analyses shown in Figure 4 did not alter the significant differences between threat+ and threat- conditions at temporal electrodes at 170 ms following stimulus onset (*t*_23_= -2.1, *p* < 0.05), parietal electrodes at 500 ms (*t*_23_= 4.2, *p* < 0.001) and motor signals at 200 ms (*t*_23_= 3, *p* < 0.01).

Furthermore, note that the neural-choice correlation analyses performed in the Results section do take participants’ responses into account, and this is now more explicitly underlined in the revised manuscript. The important advantage of the model-based approach we applied instead of a model-free split between correct and error responses is that it allows to assess not only *whether* EEG signals co-vary with behavioral responses, but also how they do so: through changes of sensitivity and/or bias.

*Finally I enjoyed the anxiety dependent analysis and the findings are clear and interesting.*

We are happy that the reviewer found our anxiety-dependent analyses of the neural effects of importance and interest.

Reviewer #3:

*[…] This part of the paper is elegant and well done, and to my knowledge, novel. The Methods section is really clear – my preference would be to see some of the explanation of the models moved into the main part of the paper, as this part, while 'technical', is really needed to understand the meaning of the subsequent analysis, and as it stands, the introduction is much less clear than the longer explanation.*

We thank the reviewer for his/her positive comments on the study. As indicated in response to the collective points above, we have attempted to be more explicit about the regression-based, model-guided methods we have applied to the neural data throughout the Results section of the revised manuscript. We hope that the reviewer would find the revised manuscript clearer in this respect.

*The final section of the paper claims a relationship with anxiety levels in individual subjects, as* threat+ *and - are differently modulate temporal areas in low anxious individuals), and* threat+ *modulates motor areas in high anxious individuals.*

*I have reservations about the second part of the dissociation in high anxious individuals mostly to do with Figure 7 – it seems to me to be the main weakness.*

We believe to have clarified and strengthened the observed anxiety-dependent dissociation between sensory and motor enhancements in response to threat+ combinations of gaze and emotion in the revised manuscript – as indicated in response to the collective points above and the specific points raised by the reviewer below.

*1) Anxiety is co-morbid with depression, and social anxiety. Were participants screened for their scores with other factors that might be correlated with anxiety?*

Participants were not screened for other dimensions such as social anxiety or depression, but only for the single dimension we had predictions on in relation to threat perception: anxiety. The reviewer is right that anxiety scores measured with the STAI questionnaire are typically correlated with depression and social anxiety scores across individuals, and we have updated the Discussion section to acknowledge the resulting limitation as to the specificity of the observed effects: “Further research should assess the specificity of these anxiety-dependent effects, in light of the growing evidence in favor of a comorbidity between anxiety and depressive disorders.”

Nevertheless, as indicated in response to the collective points above and concerning the comorbidity between depression and anxiety, it has been shown that patients with clinically defined anxiety and depression had lower mortality rates than those with depression alone (Mykletun et al., 2009). This finding thus suggests that clinical anxiety may have a specific ‘protective’ function, related to the adaptive role of anxiety in survival, which is not shared with depression. Despite their limitations (shared by most of existing studies of anxiety in the field), our findings provide empirical evidence in favor of such positive function of non-clinical anxiety in the general population.

*2) All participants bar one lie in the very centre of the normal range. In Figure 7, it looks like that 40+ score might be an outlier. R values are computed with Pearson (I assume, though I notice that isn't on the figure; it should be). If the outlying subject is excluded, does the correlation survive?*

The correlation strengths reported in the manuscript are indeed linear Pearson correlation values, and this is now specified in the text and figure legends of the revised manuscript. As indicated in response to the collective points above, we have performed several additional analyses of the data shown on Figure 7 (now Figure 8 in the revised manuscript), which should convince the reviewer of the robustness of the anxiety-dependent effects in sensory and motor regions.

First, it is important to note that the anxiety scores of our participants (between 20 and 45) fall well within the range of the normal population (< 60), and there is thus no a priori reason to exclude any of our participants solely based on his/her anxiety score. Besides, it is precisely the between-participant variability in anxiety scores that afforded their correlation with neural effects. As a first control analysis, we computed prediction intervals for the group-level regression line at the anxiety score of each participant. No participant fell outside of the 95% prediction intervals for both temporal (Figure 8) and motor areas (Figure 8) – and could thus be labeled statistically as outlier. Nevertheless, if we exclude *arbitrarily* participant #4 whose anxiety score was 45, the motor correlation with anxiety remains marginally significant (*r* = 0.35, d.f. = 21, two-tailed *p* = 0.09), as well as the interaction between state anxiety and threat level (mixed-design ANOVA, F_1,22_ = 3.1, *p* = 0.09). And within median-split high-anxious participants, the neural encoding of emotion strength in the threat+ condition remains significant at a two-tailed level (t-test against zero, *t*_10_ = 2.4, two-tailed *p* = 0.03), as well as the observed difference between threat+ and threat− conditions (paired t-test, *t*_10_ = 2.6, two-tailed *p* = 0.02).

Instead of excluding arbitrarily one participant, we performed a leave-one-out, cross-validation procedure in which we computed ‘cross-validated’ prediction intervals for the group-level regression line at the anxiety score of participant #n when the data for participant #n was excluded from the group-level regression. This procedure was repeated for the 24 participants for the anxiety-threat correlation in motor areas and showed that the neural effects of two participants fell slightly (< 20%) outside of the *cross-validated* 95% prediction intervals – including participant #4. Nevertheless, recomputing the group-level correlation in motor areas after excluding the two outliers, leaving 22 participants in the analysis, led to a significant effect (*r* = 0.48, d.f. = 20, *p* = 0.02). This leave-one-out analysis identified no outlier for the anxiety-threat correlation in temporal regions.

Together, we believe that these control analyses confirm the robustness of the anxiety-threat correlations for both temporal and motor regions, and we now describe them in the Methods section of the revised manuscript.

*Following on from that, it seems to me that there is an increase in variabilityin the 'more anxious' group, which could be driving the correlation, especially as it looks as though the difference between threat and no-threat in the high anxious group in 7B isn't significant.*

In response to this specific point, we can assure the reviewer that there is a significant difference in neural encoding in motor lateralization for the high-anxious group between threat+ and threat− conditions (paired t-test, *t*_11_ = 3.0, *p* = 0.01). Neural encoding for the threat+ condition is even significantly positive (t-test against zero, *t*_11_ = 2.6, *p* = 0.02). Note also that both tests remain significant even when participant #4 is excluded arbitrarily from the high-anxious group (see our response to the previous point). Finally, the use of a leave-one-out procedure for identifying two outliers with respect to the correlation shown on Figure 7 (now Figure 8) left the correlation significant, thereby confirming that the correlation is not driven by outlying data points.

*3) The effect in Figure 7 seems to be a weak effect at a single time point. There is no justification/explanation of the selection of either the window in 7A, or the single point in 7B (if that is what it is).*

We fully agree with the reviewer on this point. While we chose to measure the correlation with anxiety at the time point corresponding to the peak of the encoding effect in motor signals (200 ms), we chose the time window corresponding to the significant cluster for the encoding at temporal electrodes. This was indeed a discrepancy, which we have corrected by also choosing the peak of neural encoding for temporal electrodes (280 ms) to perform the correlation. We have changed accordingly the figure (previously Figure 7, now Figure 8), and the text:

“Indeed, state anxiety influenced significantly the neural encoding […] between threat+ and threat− combinations proved to be significant (Pearson correlation coefficient r = 0.51, d.f. = 22, p = 0.01; Figure 8).”

Note that both correlations with anxiety remain significant if the entire time window corresponding to the significant cluster is taken, at both temporal (170-400 ms, *r* = 0.53, d.f. = 22, *p* < 0.01) and motor electrodes (130-300 ms, *r* = 0.43, d.f. = 22, *p* < 0.05). We believe that these additional analyses further strengthen the anxiety-dependent correlations reported in the manuscript.

*4) The role of reaction time: the authors say that they found no difference in the behavioural responses between the more and less anxious groups. Did that include reaction times?*

The analyses indeed included reaction times – which do not differ between high- and low-anxious groups nor correlate with anxiety. There was no measurable difference between the two anxiety groups in terms of behavior alone. The single anxiety-dependent effect that significantly impacts behavior in our study is reported in the Results section. The strength of neural encoding of threat+ emotions at motor electrodes correlated significantly with behavioral sensitivity for high-anxious participants but not for low-anxious participants, and significantly more for high-anxious than low-anxious participants:

“Moreover, the neural encoding of threat+ emotions in motor signals correlated with behavioral sensitivity to threat+ emotions for high-anxious individuals (Pearson correlation coefficient r = 0.66, d.f. = 10, p = 0.01; Figure 8), whereas it was not the case for low-anxious individuals (Pearson correlation coefficient r = −0.42, d.f. = 10, p > 0.16, difference between coefficients, p < 0.01).”

[Editors' note: further revisions were requested prior to acceptance, as described below.]

*Figure 1 is very useful* except *I didn't understand the emotion axis in Figure 1 (filled in gray area) until I saw the bottom of Figure 3 where it is labeled as emotion sensitivity. Please label in Figure 1 and explain in the legend (in addition to the explanation in the text).*

We are sorry for not clarifying what the filled gray area is on Figure 1. The emotion axis represents the evidence to emotional expression from the most fearful face on the left, to the angriest face on the right. The filled gray area represents the difference between the two curves plotted (one for direct gaze, one for averted gaze). We added this gray area to emphasize the fact that differences between direct and averted gaze (i.e., the difference between the two curves) would be maximal for the neutral level for a change in bias between direct and averted gaze conditions, and maximal for low emotion strengths for a change in sensitivity between threat+ and threat- conditions. We now added to the figure a gray box that describes the filled gray area: 'difference direct/averted', and we added to the legend:

"Maximal effects would appear for neutral (emotionless) expressions as highlighted through the filled gray area on the emotion axis that represents the difference between the two psychometric functions for direct and averted gaze."

*How were the times in Figure 4 chosen? The times in B and C make some sense to me but I don't understand how 280 was chosen when it is so far from the region of significant differences. Also please explain why the maps in D-F don't look as though they came from the brains mapped in the middle of A-C (particularly B-C). Upon re-reading this, I think I understand the timing issue – you are mapping the deviations from 0 rather than the peak differences between the conditions. Perhaps make the emphasis on the latter equal to that on the former by making the gray boxes into bars up top to match the green and orange bars? All equally emphasized that way. Confusion regarding map alignments or lack thereof remains.*

The EEG topographies plotted in the middle of Figure 4 are taken at the latency of peak deviations from zero, and not of peak differences between conditions. While the first peak of encoding independently of conditions emerges at 280 ms, the difference between our two conditions of interest (threat+ and threat-) emerged earlier (170 ms). This is not the case for the two other peaks (500 ms and response time) as the difference between conditions corresponded to the peak deviations from zero. We plot these topographies to justify how we selected the electrodes on which we compare between the two conditions of interest – i.e., using a contrast (t-test against zero) orthogonal to the contrast between the two conditions. We now added to Figure 4 on top of these topographies: “Peak encoding across conditions”.

Moreover, as suggested by the editor, we changed the gray boxes to gray bars on top of the green and orange bars to equally emphasize the deviations from zero and the differences between conditions (revised Figure 4). Yet, our main goal was to address the cognitive and neural differences between our conditions of interest (threat+ and threat-) rather than studying the encoding of emotion per se. We therefore still believe that it might be useful to differentiate visually these two types of statistical differences with a stronger emphasis on the difference between threat+ and threat- conditions.

The brain maps plotted in Figure 4 represent the difference in encoding between threat+ and threat- conditions at the source level, at the times where there was a significant difference between these conditions (gray bar). We now added on Figure 4 on top of these brain maps: “Source of encoding difference T+/T-”. Source reconstruction analyses localize the regions of the brain that generate the scalp-recorded EEG signals. Similar averaged scalp-recorded EEG signals can therefore stem from different brain sources. Accordingly, temporal EEG components (N170) were shown to have different brain sources, including the fusiform gyrus (e.g. Itier and Taylor, 2002) and the superior temporal sulcus (e.g. Batty and Taylor, 2003). Here we report both these regions as sources of the difference between threat+ and threat- encoding. Also, brain sources of parietal EEG activity (P300) include widely separated brain regions within which frontal and temporal regions (Nieuwenhuis et al., 2005), similarly to those reported in the present study when computing the sources of difference between threat+ and threat- related to parietal scalp-recorded EEG. Finally, revealing dorsal central motor-related regions in f (related to the topography in c at response time), and not e (related to the topography in b at 500 ms) is not surprising at response time.

We also added this sentence to the Results section:

“To assess which brain regions generated the scalp-recorded EEG signals, we computed the cortical sources of this enhanced encoding of threat-signaling emotions by performing the same regression approach to minimum-norm current estimates distributed across the cortical surface.”

*For the general reader, could you add a word of background about the lateralization difference illustrated in Figure 6 and discussed in the subsection “Early neural encoding of threat-signaling emotions in motor preparation”? It appears to be a robust difference – is it known and expected? Surprising in any way? Remember that the* eLife *readership is broad and not necessarily in the know about much intra-field background.*

Following the editor's advice, we added some background about the lateralization difference illustrated in Figure 6, in the subsection “Early neural encoding of threat-signaling emotions in motor preparation”:

"Limb movement execution and preparation coincide with suppression of low-frequency (8-32 Hz) activity that is stronger in the motor cortex contralateral as compared to ipsilateral to the movement. Thus, subtracting the contralateral from ipsilateral motor cortex activity is expected to result in a positive measure of motor preparation".